# Learning dynamics in linear recurrent neural networks

**Alexandra M. Proca** [1]   **Clémentine C.J. Dominé** [2]   **Murray Shanahan** [1]   **Pedro A.M. Mediano** [1 3]

## Abstract

Recurrent neural networks (RNNs) are powerful models used widely in both machine learning and neuroscience to learn tasks with temporal dependencies and to model neural dynamics. However, despite significant advancements in the theory of RNNs, there is still limited understanding of their learning process and the impact of the temporal structure of data. Here, we bridge this gap by analyzing the learning dynamics of linear RNNs (LRNNs) analytically, enabled by a novel framework that accounts for task dynamics. Our mathematical result reveals four key properties of LRNNs: (1) Learning of data singular values is ordered by both scale and temporal precedence, such that singular values that are larger and occur later are learned faster. (2) Task dynamics impact solution stability and extrapolation ability. (3) The loss function contains an effective regularization term that incentivizes small weights and mediates a tradeoff between recurrent and feedforward computation. (4) Recurrence encourages feature learning, as shown through a novel derivation of the neural tangent kernel for finite-width LRNNs. As a final proof-of-concept, we apply our theoretical framework to explain the behavior of LRNNs performing sensory integration tasks. Our work provides a first analytical treatment of the relationship between the temporal dependencies in tasks and learning dynamics in LRNNs, building a foundation for understanding how complex dynamic behavior emerges in cognitive models.

[1]Department of Computing, Imperial College London, London, United Kingdom [2]Gatsby Computational Neuroscience Unit, University College London, London, United Kingdom [3]Division of Psychology and Language Sciences, University College London, London, United Kingdom. Correspondence to: Alexandra M. Proca <a.proca22@imperial.ac.uk>.

*Proceedings of the 42nd International Conference on Machine Learning*, Vancouver, Canada. PMLR 267, 2025. Copyright 2025 by the author(s).

## 1. Introduction

Recurrent neural networks (RNNs) are important tools in both machine learning and neuroscience for learning tasks with temporal dependencies. Recently, (linear) recurrent architectures (state space models), have had a resurgence of popularity in long-range sequence modeling (Gu et al., 2020; 2022; Orvieto et al., 2023; Gu & Dao, 2024). In tandem with the success of dynamical systems theory in describing neural activity related to motor control, working memory, and decision-making (Remington et al., 2018a; Vyas et al., 2020; Khona & Fiete, 2021), RNNs have also become a popular choice for cognitive models of neural dynamics (Barak, 2017), as they not only replicate recurrent dynamics recorded in animals but are also capable of performing abstractions of the same cognitive tasks used in experiments (Mante et al., 2013; Engel et al., 2015; Chaisangmongkon et al., 2017; Wang et al., 2017; Masse et al., 2018; Remington et al., 2018b; Orhan & Ma, 2018; Masse et al., 2020; Beirán et al., 2023). More generally, RNNs present an interesting model of study due to the complex computational capabilities given by their hidden layer that evolves with time and which is a universal approximator of any open dynamical system (Doya, 1993; Schäfer & Zimmermann, 2007).

Accompanying the popularity of RNNs, there have been significant efforts dedicated to their theoretical understanding, both from deep learning theoreticians (Cohen-Karlik et al., 2023; Orvieto et al., 2024; Zucchet & Orvieto, 2024) and neuroscientists relating these findings to observations about the brain (Sussillo & Barak, 2013; Mastrogiuseppe & Ostojic, 2018; Yang et al., 2019; Schuessler et al., 2020a;b; Turner et al., 2021; Dubreuil et al., 2022; Farrell et al., 2022; Turner & Barak, 2023; Driscoll et al., 2024; Liu et al., 2024). However, most theoretical studies of RNNs are done at the end of training — analyzing properties of the solutions they find, but ignoring the learning process itself (Saxe et al., 2020). Of the work that does study learning, the focus is often related to practical considerations about training, such as learning long-range dependencies. Overall, despite the widespread use and known complex computational abilities of RNNs, it is still unknown how their underlying functional structures emerge as a result of training on temporally-structured tasks.

One related line of previous work has focused on using deep linear networks to analyze learning dynamics (Saxe et al., 2014; 2018; Braun et al., 2022; Dominé et al., 2025). Although unable to solve nonlinear problems (note however that there has been progress to overcome this limitation (Saxe et al., 2022; Sandbrink et al., 2024)), these networks exhibit complex nonlinear learning dynamics and are analytically tractable, providing a useful framework for theoretical investigation. Applied to cognitive neuroscience, the study of learning dynamics has been used to propose a theory of semantic development (Saxe et al., 2018), cognitive flexibility (Sandbrink et al., 2024), and localization in receptive fields (Lufkin et al., 2024), among other work. Despite its successes, the analytical treatment of learning dynamics in linear networks has primarily remained in the domain of feedforward networks. In order to more broadly characterize learning, however, theory needs to account for the impact of dynamic task settings and the rich structure endowed by recurrent networks, especially since it is such a critical component of neural computation. Of the few prior studies of learning dynamics in linear RNNs, Schuessler et al. (2020b) showed that networks make low-rank changes to their connectivity during learning and Smékal et al. (2024) showed how overparameterization accelerates convergence time by studying the frequency domain. However, the influence of temporally structured data on learning has not been studied analytically to our knowledge.

In this work, we study the learning dynamics of linear RNNs (LRNN) to better understand the influence of temporal data on learning in recurrent cognitive systems, unifying the areas of RNN theory and learning dynamics. Our theoretical results contribute to explanations of many phenomena spanning both topics, including low-rank connectivity (Mastrogiuseppe & Ostojic, 2018), rich and lazy learning (Farrell et al., 2023), extrapolation capabilities (Cohen-Karlik et al., 2023), and network stability (Sompolinsky et al., 1988). Taken together, this represents a substantial step towards the theoretical understanding of learning in recurrent deep learning models, building a foundation for new theories and hypotheses of learning in neural networks and the brain.

Our contributions are as follows:

- We provide, for the first time, a closed-form analytical expression for the energy function of LRNNs decoupled along singular/eigen-value dimensions. We use this result, together with a novel framework to describe task dynamics, to accurately predict solutions found by LRNNs.

- We identify how both the magnitude and temporal ordering of data singular values affect learning speed.

- We describe how task dynamics impact network stability and extrapolation ability, even in cases where the

network achieves 0 loss.

- We identify an effective regularization term in the energy function that incentivizes small weights, and a phase transition in the connectivity modes that leads to low-rank solutions.

- We derive the neural tangent kernel (Jacot et al., 2018) for finite-width LRNNs and show that recurrence facilitates feature learning.

- We demonstrate the generalizability of our results by applying our theoretical framework to describe the behavior of LRNNs trained on sensory integration tasks, relaxing our prior assumptions.

## 2. Mathematical setup

### 2.1. Model

We study a LRNN (Figure 1) parameterized by matrices $W_x \in \mathbb{R}^{N_h \times N_x}, W_h \in \mathbb{R}^{N_h \times N_h}, W_y \in \mathbb{R}^{N_y \times N_h}$ with a hidden state $\boldsymbol{h}_t \in \mathbb{R}^{N_h}$ that receives an input $\boldsymbol{x}_t \in \mathbb{R}^{N_x}$ at each timestep $t$ and updates its hidden state. For simplicity, in the main text we study the *single-output* case, where the network only produces an output $\hat{\boldsymbol{y}}_T \in \mathbb{R}^{N_y}$ at the last timestep $T$. In Appendix M, we generalize our approach to networks trained to produce outputs $\hat{\boldsymbol{y}}_t$ at every timestep $t$ (the autoregressive *T-output* case). The network is characterized by the equations

$$\boldsymbol{h}_{t+1} = W_h \boldsymbol{h}_t + W_x \boldsymbol{x}_t \,, \tag{1}$$
$$\hat{\boldsymbol{y}}_T = W_y \boldsymbol{h}_{T+1} \,. \tag{2}$$

We initialize the hidden state $\boldsymbol{h}_1$ as a vector of zeros, yielding

$$\boldsymbol{h}_{t+1} = \sum_{i=1}^{t} W_h^{t-i} W_x \boldsymbol{x}_i \,. \tag{3}$$

We analyze learning in the LRNN when trained using backpropagation through time on the squared error over $P$ trajectories $\{\boldsymbol{x}_{p,1}, \boldsymbol{x}_{p,2}, \ldots, \boldsymbol{x}_{p,T}, \boldsymbol{y}_{p,T}\}_{p=1}^{P}$

$$\mathcal{L} = \frac{1}{2} \sum_{p=1}^{P} \|\boldsymbol{y}_{p,T} - W_y (\sum_{i=1}^{T} W_h^{T-i} W_x \boldsymbol{x}_{p,i})\|^2 \tag{4}$$

### 2.2. Temporal singular values

With the model and loss function fixed, our next step is to specify a task for the model to learn. In this linear setting, the task is fully specified by the sequence of matrices $\Sigma^{YX_t} = \sum_{p=1}^{P} \boldsymbol{y}_{p,T} \boldsymbol{x}_{p,t}^{\top}$, the input-output correlation matrix between the input $\boldsymbol{x}_{p,t}$ at timestep $t$ and the final output

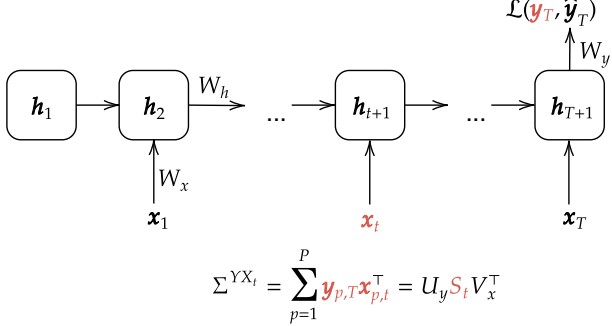

$$\Sigma^{YX_t} = \sum_{p=1}^{P} \boldsymbol{y}_{p,T} \boldsymbol{x}_{p,t}^{\top} = U_y S_t V_x^{\top}$$

*Figure 1.* **Linear RNN model captures task dynamics through temporally-dependent singular values.** The data correlation matrices $\Sigma^{YX_t}$ have constant left and right singular vectors, varying only in their singular values $S_t$ across time.

target $\boldsymbol{y}_{p,T}$. Extending the approach by Saxe et al. (2014; 2018), we can represent these matrices through either their singular value decomposition (SVD; case 1 in the assumptions below), or their eigendecomposition (case 2), which enables us to account for task dynamics. In the main text, we follow the precedence from prior work and base our analysis around a derivation using SVD, which, although more restrictive with task dynamics, simplifies the setting and allows for non-square networks. Then, in Appendices N and O, we use an eigendecomposition to derive a similar but more general form (accounting for complex eigenvalues thus allowing for rotational dynamics) and show that our framework and results extend naturally to this case. To simplify our derivations, we make the following assumptions:

**Assumption 1 (whitened input):** Inputs are uncorrelated and whitened across all timesteps and dimensions, such that $\Sigma^{X_t X_t} = I; \Sigma^{X_t X_{t'}} = \boldsymbol{0}, t \neq t'$.

**Assumption 2 (constant singular vectors *or* eigenvectors):** All input-output correlation matrices have either (1) constant left and right singular matrices $U_y, V_x$ and only vary in their singular values $S_t$ over a trajectory, such that $\Sigma^{YX_t} = U_y S_t V_x^{\top}, \forall t$; or (2) constant eigenvectors $P$ and only vary in their eigenvalues $D$ over a trajectory, such that $\Sigma^{YX_t} = P D_t P^{\dagger}$.

**Assumption 3 (aligned model):** The model is aligned to either (1) the data singular vectors at initialization such that $U_y^{\top} W_y(0) R_y, R_y^{\top} W_h R_x, R_x^{\top} W_x(0) V_x$ yield diagonal matrices $\overline{W}_y(0), \overline{W}_h(0), \overline{W}_x(0)$ for some orthogonal matrices $R_y, R_x$, or (2) the data eigenvectors $P$ such that $P^{\dagger} W_y(0) P, P^{\dagger} W_h(0) P, P^{\dagger} W_x(0) P$ yield diagonal matrices $\overline{W}_y(0), \overline{W}_h(0), \overline{W}_x(0)$.

Although seemingly restrictive, we argue these assumptions still capture meaningful learning scenarios. For example: (1) data can be whitened using the innovations form of a Kalman filter (Durbin & Koopman, 2012); (2)

singular/eigen-vectors are constant if data is generated by a diagonalizable LRNN teacher (Appendices B and N), and in fact, because we don't restrict the dynamics of the singular/eigen-values, our form captures more general settings than the standard teacher-student setup (which constrains task dynamics to the form $\delta\lambda^{T-t}$); (3) prior work has shown that model alignment occurs early in training for networks initialized with small random weights (Atanasov et al., 2022) and there are theoretical and practical justifications for diagonalizable state spaces (Hazan et al., 2018; Gupta & Berant, 2022).

## 3. Results

### 3.1. LRNN energy function

With the aforementioned assumptions, we can diagonalize the network, eliminating cross-terms. Let $a_{\alpha}, b_{\alpha}, c_{\alpha}$ be the $\alpha^{\text{th}}$ diagonal entry of $\overline{W}_x, \overline{W}_h, \overline{W}_y$, respectively, and $s_{\alpha,t}$ be the $\alpha^{\text{th}}$ singular value (SV) of $S_t$. Assuming a small learning rate $1/\tau$ (i.e., the *gradient flow* regime), we can write the gradients of the network parameters as a set of differential equations in terms of these variables, or *connectivity modes*, the dynamics of which decouple across SV dimensions $\alpha$ (Appendix B). We refer to $a_{\alpha}$ as the input, $b_{\alpha}$ the recurrent, and $c_{\alpha}$ the output connectivity mode. Their dynamics are given by

$$\tau \frac{d}{dt_{\theta}} a_{\alpha} = \sum_{i=1}^{T} c_{\alpha} b_{\alpha}^{T-i} (s_{\alpha,i} - c_{\alpha} b_{\alpha}^{T-i} a_{\alpha}) \qquad (5)$$

$$\tau \frac{d}{dt_{\theta}} b_{\alpha} = \sum_{i=1}^{T-1} (T-i) c_{\alpha} b_{\alpha}^{T-i-1} a_{\alpha} (s_{\alpha,i} - c_{\alpha} b_{\alpha}^{T-i} a_{\alpha}) \qquad (6)$$

$$\tau \frac{d}{dt_{\theta}} c_{\alpha} = \sum_{i=1}^{T} b_{\alpha}^{T-i} a_{\alpha} (s_{\alpha,i} - c_{\alpha} b_{\alpha}^{T-i} a_{\alpha}) , \qquad (7)$$

where $t_{\theta}$ refers to timesteps of gradient-based learning as opposed to the trajectory timesteps $t$. Our first result shows that these dynamics arise from gradient descent on an energy function.

**Lemma 3.1.** *Given Assumptions 1-3, the energy function of the LRNN is given by*

$$E(a_{\alpha}, b_{\alpha}, c_{\alpha}) = \frac{1}{2\tau} \sum_{i=1}^{T} (s_{\alpha,i} - c_{\alpha} b_{\alpha}^{T-i} a_{\alpha})^2 . \qquad (8)$$

To ease notation, we omit specifying $\alpha$ when referring to connectivity modes in the remainder of the paper, although note that all terms $(s_t, a, b, c)$ still refer to a particular SV dimension $\alpha$. We also generally refer to the input-output modes $(ac)$ together and treat them as a single term since there isn't any meaningful distinction between them.

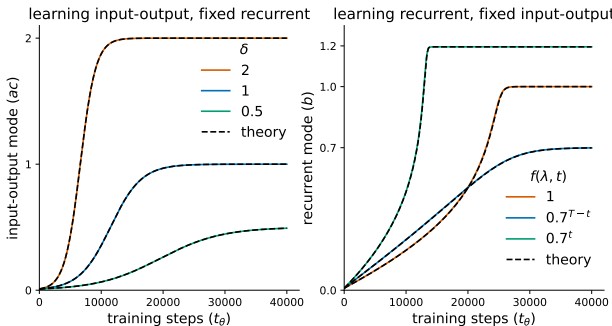

*Figure 2.* **Learning dynamics of input-output and recurrent connectivity modes in a linear RNN.** Data singular values $s_t$ can be decomposed into a scaling term $\delta$ and temporal term $f(\lambda, t)$. (*Left*) The input-output modes learn the scaling component, while (*right*) recurrent modes learn the dynamic component. The colored lines are simulations and dashed lines are the corresponding theoretical predictions.

To provide some intuition about the reduced form, the magnitude of the data SVs ($s_{\alpha,t}$) correspond to the strength of correlation between the input ($\boldsymbol{x}_t$) at trajectory timestep $t$ and the output target ($\boldsymbol{y}_T$) in different SV dimensions $\alpha$. In this work, we're interested in understanding how recurrence and the task dynamics (given by $s_{1:T}$) impacts the LRNN's learning dynamics.

### 3.2. Solutions to LRNN learning dynamics

While perhaps trivial, we can think of the LRNN as performing two functions: the input-output mode ($ac$) performs a constant scaling and the recurrent mode ($b$) learns a time-dependent function. Thus, by decomposing the data SVs ($s_{1:T}$) into a constant and temporal component, we might better understand the solutions LRNNs converge to. We decompose each data SV as $s_t = \delta f(\lambda, t)$, where $\delta$ is constant across all data SVs, and $f(\lambda, t)$ is some function parameterized by $\lambda$ that is dependent on trajectory timestep $t$.

We derive a full solution for the learning dynamics of the input-output modes when recurrent modes are frozen (Appendix C), as well as a local approximation to the recurrent modes when input-output modes are frozen (Appendix D). Intuitively, the network should use recurrent modes to learn the dynamic component of the data since it varies in its contribution to the output through time, whereas input-output modes do not vary with time and thus can only contribute some form of scaling. By studying the learning dynamics of the recurrent and input-output modes separately, we confirm that they indeed learn these different components (Figure 2).

The distinction between dynamic and scaling components of data SVs also highlights an important difference in learning dynamics between (deep) feedforward and recurrent linear networks. Feedforward linear networks learn the largest SVs first. In recurrent networks, however, the loss is computed

over $T$ SVs in each dimension (as opposed to one) and the network must optimize for SVs across time. A consequence is that SVs at different timesteps are weighted differently in the gradient. SV trajectories that are larger and have SVs occurring *later* in the trajectory are learned faster (assuming recurrent connectivity modes are initialized $b < 1$). We can see this effect by looking at the gradients of the connectivity modes (Equations (5) to (7)): gradients from early trajectory timesteps are weighted by the recurrent mode $b$ exponentially with trajectory length. Since we initialize connectivity modes to be less than 1, this has the effect of downscaling the gradient contribution from earlier trajectory timesteps compared to later timesteps. Thus, we see a more complex portrait of the effect of both SV magnitude and SV dynamics (time) playing into the ordering of learning in recurrent networks. This effect can be seen in Figure 2 (right) – the blue curve converges to a smaller solution than the orange curve but is initially learned faster, which differs from the behavior of feedforward networks (e.g., the left plot), and is driven by the fact that the singular value at the last timestep is larger for the blue curve than the orange curve (from $\delta$). We study this further in Appendix E.

### 3.3. Task dynamics determine solution stability and extrapolation ability

RNNs suffer from problems related to stability during training and inference. By stability we refer to the state of the RNN parameters which may lead to exploding gradients or diverging hidden layer activity. Because of the exponential effect of the recurrent layer, (nonlinear) RNNs with eigenvalues larger than 1 exhibit chaotic behavior (Sompolinsky et al., 1988). By looking at the energy function (Equation (8)), we can further see the well-known effect of vanishing (exploding) gradients, given by $|b| < 1$ ($|b| > 1$) as $T \to \infty$ (Bengio et al., 1994; Hochreiter et al., 2001; Pascanu et al., 2012), which makes training on tasks with long-range dependencies challenging and to which there have been numerous methods introduced to alleviate these difficulties (Hochreiter & Schmidhuber, 1997; Le et al., 2015; Orvieto et al., 2023; Zucchet et al., 2023). Another open problem in RNNs is their ability to extrapolate (or interpolate) to sequence lengths that differ from those trained on, which is not well understood (Cohen-Karlik et al., 2023; Beirán et al., 2023). Here, we study how network stability and extrapolation ability are impacted by an additional factor: the underlying task dynamics a RNN is trained on.

To do this, we study task dynamics that are perfectly learnable (provably the only task dynamics with 0-loss solutions; Appendix F), but differ in their hidden layer stability or ability to extrapolate. Recall that we can decompose data SVs as $s_t = \delta f(\lambda, t)$. We distinguish between three cases with known analytical 0-loss solutions (Appendix F.3), which offer natural settings to study how perfectly-learnable data

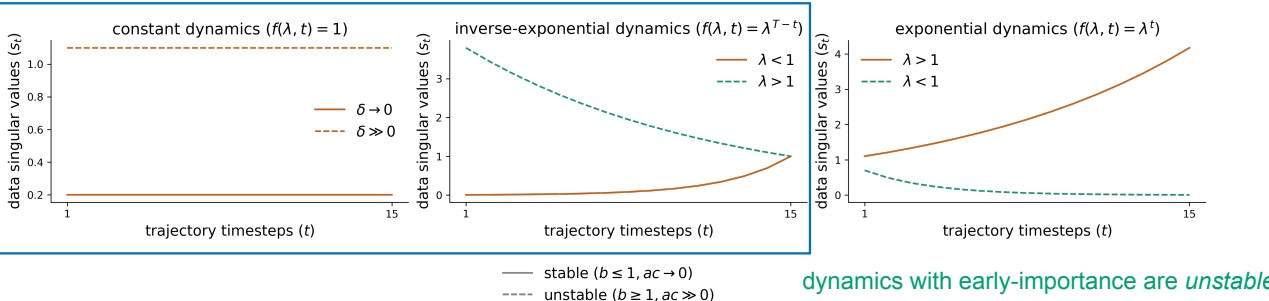

*Figure 3.* **Task dynamics determine solution stability and extrapolation ability.** For RNNs trained on (*left*) constant dynamics ($f(\lambda, t) = 1$), solution stability is dependent on the scaling term $\delta$, where $\delta \to 0$ is stable and $\delta \gg 0$ is less stable. (*Middle*) Inverse-exponential ($f(\lambda, t) = \lambda^{T-t}$) and (*right*) exponential ($f(\lambda, t) = \lambda^t$) dynamics produce unstable solutions when $\lambda > 1$ and $\lambda < 1$, respectively, which correspond to *early-importance* dynamics ($s_t > s_{t+1}$). Further, the solution to the input-output modes ($ac = \delta\lambda^T$) for exponential dynamics depends on trajectory length, so solutions learned for one length will not extrapolate to other trajectory lengths.

impacts solution stability and extrapolation ability. We consider cases where the data SVs are *constant* ($f(\lambda, t) = 1$), change *inverse-exponentially* ($f(\lambda, t) = \lambda^{T-t}$), or change *exponentially* ($f(\lambda, t) = \lambda^t$). By varying $\delta, \lambda$, we can parameterize the task dynamics differently and elicit particular network behavior. We note that technically all of these dynamics can be reparameterized as inverse-exponential dynamics when the trajectory length is fixed (Appendix F.1), but for simplicity, we will keep these separate.

For constant task dynamics, the global solution exists at $b = 1, ac = \delta$; this can be understood as 'equally weighting' the input at each timestep, while the input-output connectivity modes learn an appropriate scaling $\delta$. For inverse-exponential task dynamics, the minimum is found at $b = \lambda, ac = \delta$, as the dynamics of the singular values ($s_t = \delta f(\lambda, t), \ f(\lambda, t) = \lambda^{T-t}$) correspond exactly to the dynamics of the LRNN ($cb^{T-t}a$). Finally, for exponential task dynamics, the minimum exists at $b = 1/\lambda, ac = \delta\lambda^T$.

By studying these solutions, we can first observe that inverse-exponential and exponential task dynamics yield *unstable* solutions (where the recurrent mode $b > 1$) for $\lambda > 1$ and $\lambda < 1$, respectively (green dashed lines in Figure 3). In both cases, the data SVs decrease across the trajectory ($s_t > s_{t+1}$); we hence refer to this as *early-importance*. Due to the instability of exploding gradients as the recurrent mode $b$ increases over 1, early-importance dynamics are more challenging, if not impossible, to learn as trajectory length increases ($T \to \infty$).

Constant dynamics are essentially an intermediary between exponential and inverse-exponential dynamics (i.e., because $f(\lambda = 1, t) = 1^t = 1^{T-t} = 1$). Constant dynamics are common, as they correspond to basic integration of input. They also present a way to study the influence of the scaling term ($\delta$) on solution stability. Constant dynamics have stable solutions when the scaling term is small ($\delta \to 0$) because it keeps SVs and input-output modes small as the recurrent

mode ($b$) approaches 1; however, when the scaling term is large ($\delta \gg 0$), optimization is more challenging as solutions approach unstable solutions near $b = 1$ (left in Figure 3). This observation about $\delta$ also generalizes to other dynamics when the solution for the recurrent mode is not close to 0 ($b \gg 0$).

Although each case of task dynamics we consider here has a 0-loss solution, not all of these solutions extrapolate perfectly to other trajectory lengths $T$. In particular, the global solution for exponential task dynamics is dependent on the trajectory length $T$ ($ac = \delta\lambda^T$). As such, this solution will *not* perfectly extrapolate to trajectory lengths that differ to the one trained on, and in fact the error will grow as the difference in trajectory length increases (see Appendix F.1).

In conclusion, we can see that even for data that is perfectly learnable, properties of the task dynamics crucially impact the stability of solutions and the ability to extrapolate to other trajectory lengths. In particular, we find that (1) data with correlations that decrease over trajectory time produce unstable solutions, (2) task dynamics with a large scaling term ($\delta$) are less stable, and (3) task dynamics with solutions that depend on trajectory length (such as exponential dynamics) do not extrapolate with 0-loss to other trajectory lengths. We show in Section 3.6 and Appendix L that these findings hold for RNNs without our theoretical assumptions.

### 3.4. Connectivity modes exhibit phase transitions between recurrent and feedforward computations

In the previous section we studied task dynamics with perfect solutions. However, most real-world tasks will, naturally, not exhibit inverse-exponential dynamics and may not have perfect 0-loss solutions (although the loss may be low in practice). A separate observation is that RNNs initialized with small random weights seem to learn low-rank solutions along effective 'task dimensions' (Schuessler et al., 2020b;

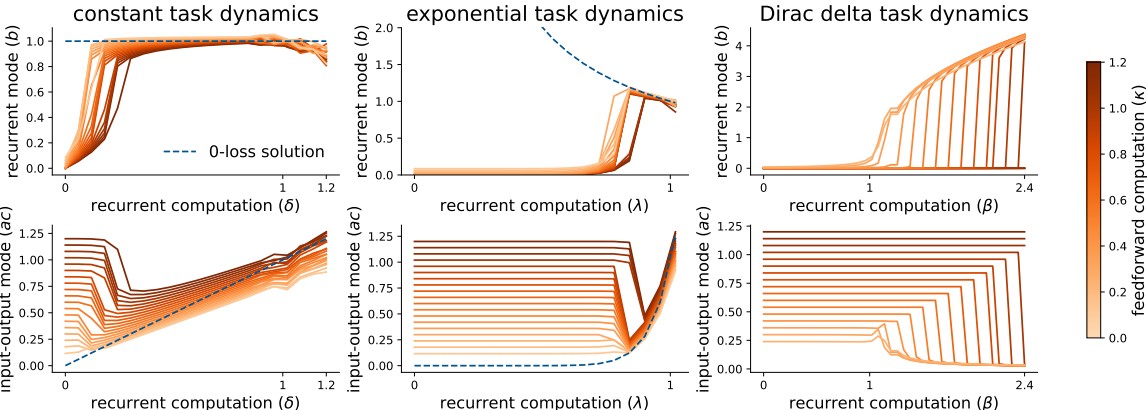

*Figure 4.* **Task dynamics modulate phase transitions of network connectivity modes.** We plot the learned solution of the connectivity modes for (*left*) constant, (*middle*) exponential, and (*right*) Dirac delta task dynamics. When the recurrent computation of the task dynamics (*x-axis*) is small, (*top*) recurrent modes ($b$) remain near 0 and (*bottom*) input-output modes ($ac$) learn the feedforward computation ($\kappa$; indicated by color). As the recurrent computation increases, the network transitions to instead learn the recurrent computation (dotted line is the 0-loss solution), manifesting as a phase transition in both the recurrent and input-output connectivity modes.

Liu et al., 2024). This motivates us to ask a question that is two-fold: (1) how task dynamics impact what a LRNN learns when it cannot perfectly fit (all of) the data, and (2) when pruning connectivity modes (and hence low-rank solutions) may be a useful strategy. To gain intuition, we can rewrite the energy function as a sum of two terms.

**Lemma 3.2.** *The energy function of a LRNN can be decomposed into a data-driven term and an effective regularization term given by*

$$E = \underbrace{\frac{1}{2\tau}\left(\sum_{i=1}^{T} s_i^2 - 2s_i cb^{T-i}a\right)}_{\text{data-driven term}} + \underbrace{\frac{1}{2\tau}c^2a^2\frac{1-b^{2T}}{1-b^2}}_{\text{effective regularization term}} .$$ (9)

*where, as* $T \to \infty$, *the second term goes to infinity for* $cba \gg 0$.

For task dynamics that are learnable with low loss, both terms cancel out. However, when the LRNN cannot fit the data, the second term acts as an effective regularizer that incentivizes connectivity modes to remain close to 0 (Appendix G). This suggests that LRNNs might have an implicit bias towards effectively low-rank solutions, both when tasks span only a few dimensions (i.e., $s_{\alpha,t} \approx 0, \forall t$ for most $\alpha$) and when tasks are not perfectly learnable.

To further investigate how data impacts what a LRNN learns, we modify the task dynamics we studied earlier (constant, exponential, inverse-exponential) to have a SV at the last timestep, $s_T$, that *does not* follow the task dynamics as the rest of the trajectory. More specifically,

$$s_t = \begin{cases} \delta f(\lambda, t) & \text{if } t < T \\ \kappa & \text{if } t = T \end{cases}$$ (10)

where $\kappa \neq \delta f(\lambda, T)$. Here, there is no way for the network to perfectly fit the task dynamics (which would only be the case if $\kappa = \delta f(\lambda, T)$). We can think of the two cases as a recurrent contribution to the task dynamics ($t < T$) and a feedforward contribution ($t = T$).

Using this setup, we experiment with varying each of the different parameters to show that each one can affect the underlying task dynamics and consequently influence the network's behavior. For example, when studying constant task dynamics (where $\lambda = 1$ by default), we change the value of the scaling term ($\delta$). Instead, when studying exponential task dynamics, we change the dynamic term ($\lambda$).

Finally, in all settings, we vary the feedforward computation ($\kappa$) independently of the other (recurrent) parameters, to study how the network deals with the tradeoff between learning solutions for $t < T$ and $t = T$, which cannot be learned simultaneously. In particular, this construction forces the network to either approximate the recurrent dynamics (constant: $ac \to \delta, b \to \lambda$; exponential: $ac \to \delta\lambda^T, b \to 1/\lambda$) or the feedforward computation ($ac \to \kappa, b \to 0$), each of which will incur a non-zero error from its counterpart.

We run simulations varying across these different task dynamics and plot the final solutions the connectivity modes converge to (Figure 4). We find that the aforementioned tradeoff between recurrent and feedforward computation manifests as a rapid phase transition of connectivity mode values across different task dynamics and becomes sharper as trajectory length $T$ increases. We show that this phase transition can be induced by varying either the scaling term ($\delta$; left in Figure 4) or the dynamic term ($\lambda$; middle in Figure 4). When the error term is dominated by the feedforward computation ($\kappa$ is large, $\delta f(\lambda, t)$ is small), the network effectively prunes the recurrent mode ($b \to 0$) rather

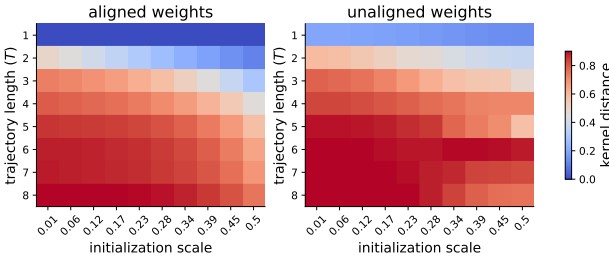

*Figure 5.* **Recurrence encourages feature learning.** Phase plots illustrating the kernel distance of the NTK from initialization as a function of trajectory length and initialization scale for LRNN initialized with weights that are (*left*) aligned and (*right*) unaligned.

than approximating the recurrent computation ($s_{1:T-1}$) and the input-output mode learns the feedforward computation ($ac \approx \kappa$). As $\delta f(\lambda, t)$ increases and the recurrent computation becomes more important (has a greater contribution to the loss), the network rapidly transitions to a regime where it approximates the task dynamics and approaches the 0-loss solution for the case where $\kappa = \delta f(\lambda, T)$, following the dynamic trajectory to the last timestep (dashed line in Figure 4) and ignoring the feedforward computation.

To further illustrate the tradeoff between feedforward and recurrent computation, we simulate networks trained on task dynamics produced by two Dirac delta functions, which have no 0-loss solution:

$$s_t = \begin{cases} \beta & \text{if } t = 1 \\ \kappa & \text{if } t = T \\ 0 & \text{otherwise} \end{cases} \tag{11}$$

As before, we vary the recurrent and feedforward computations separately by independently changing $\beta$ and $\kappa$. As shown in Figure 4 (right), we again see a sharp transition as the recurrent computation ($\beta$) increases, where the recurrent mode becomes non-zero, while simultaneously, the input-output mode decreases in magnitude. In Appendix H, we show that the phase transition depends only on the ratio of recurrent to feedforward computation ($\beta/\kappa$), such that the recurrent computation is pruned when this ratio is small. Using Landau theory, we show analytically that this corresponds to a first-order phrase transition for $T > 3$.

Taken altogether, these results suggest an implicit bias towards small weights and low-rank connectivity in RNNs, mediated by an effective regularization term. They further illustrate a tradeoff between feedforward and recurrent computations, and show cases where the network prunes connectivity modes to deal with task dynamics that are not perfectly learnable, leading to low-rank connectivity. If the recurrent part of the computation is small, and/or there is a strong correlation with the input at the final timestep and the output, the network will prune that dimension, leading to a low-rank RNN. While this behavior might seem an artifact

of the setting, we emphasize that the cases we study here are likely not the only task dynamics with representational tradeoffs when there are $T$ singular values to fit to in a single dimension. It's unclear how RNNs might prioritize learning certain computations over others in various scenarios. These results, together with those on learning speed (Section 3.2), suggest a recency bias, although the cumulative effect of the recurrent computation can outweigh this.

### 3.5. Recurrence facilitates rich learning

Prior work has identified two distinct learning regimes in neural networks: feature learning (*rich* learning), where networks learn structured task-relevant representations, and non-feature learning (*lazy* learning), where networks perform high-dimensional projections of the input (Heij et al., 2007; Yang, 2020; Farrell et al., 2023); rich learning typically occurs in networks initialized with small random weights and lazy in networks with large weights. Significant progress has been made in the theoretical understanding of these regimes, particularly in feedforward architectures (Arora et al., 2019; Azulay et al., 2021; Braun et al., 2022; Saxe et al., 2022; Kunin et al., 2024; Dominé et al., 2025). However, research into how non-feedforward architectures affect these learning regimes remain limited. Notably, Liu et al. (2024) examined the role of weight connectivity in shaping learning regimes in RNNs and Schuessler et al. (2024) showed that RNNs have different learning regimes characterized by either aligned or oblique recurrent dynamics. Building on this line of inquiry, we explore how recurrence impacts feature learning dynamics. Specifically, we investigate whether recurrent architectures impose additional constraints on the learning problem, thereby biasing the network towards the rich learning regime.

The rich and lazy learning regimes are typically evaluated using the *neural tangent kernel* (NTK) (Jacot et al., 2018), which is constant during lazy learning and non-constant during rich learning. We derive the NTK for finite-width LRNNs (Appendix I), which we then use to study what learning regimes emerge in LRNNs with different initializations and trajectory lengths. Importantly, our derivation does *not* place any assumptions on the alignment of LRNN weights as in the prior sections.

To quantify feature learning, we measure the kernel distance between the NTK at initialization and the end of training for LRNNs trained on constant task dynamics as a function of trajectory length and weight initialization scale in both the aligned and unaligned case (Figure 5). As expected, we see that the kernel moves further in networks with smaller initializations relative to the target ($= 1$), but surprisingly, the NTK still moves substantially even across larger initializations (Appendix J). We also find that the kernel distance increases as the network transitions from a feedforward net-

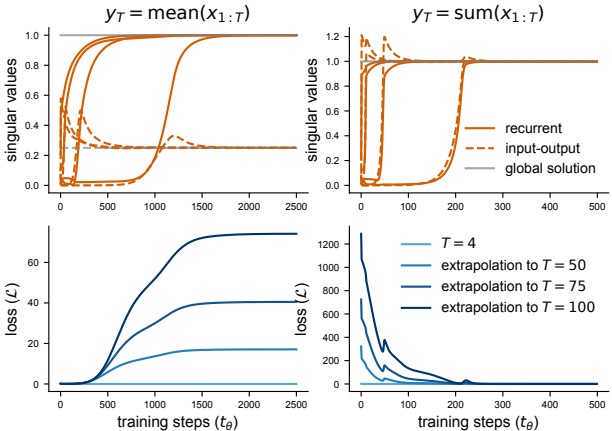

*Figure 6.* **Theory predicts learned solutions and extrapolation ability in sensory-integration task.** We train LRNNs on either producing an output that is the (*left*) mean of the input in each dimension, or the (*right*) sum. (*Top*) The LRNN singular values converge to our theoretical predictions for constant task dynamics. (*Bottom*) We accurately predict extrapolation ability for sum-integration, and lack thereof for mean-integration.

work (i.e., $T = 1$) to a recurrent network, indicating greater feature learning. In Appendix J, we further study the effects of rotational tasks, larger initializations and widths, and independently-initialized modes.

### 3.6. Sensory integration task

Although in this work we study a simplified setting that cannot fully capture all of the rich neural dynamics exhibited in animals, LRNNs can still learn and perform some basic computations studied in neuroscience, such as temporal integration of input and rotational dynamics (Khona & Fiete, 2021). Here, as a proof of concept, we study a sensory integration task where we remove our prior assumptions about whitened data and aligned weights, and show that the insights developed in our theoretical model generalize to predict behavior in this setting.

We consider two versions of a sensory integration task where the network is given noisy input in several dimensions and tasked with producing either the mean input activity in each dimension, or the sum of the input in each dimension. We train LRNNs with small random weights, making no other architectural assumptions. In such a task, the output is equally correlated across inputs at all trajectory timesteps, thus exhibiting *constant* dynamics. Our theory predicts that networks trained on tasks with constant dynamics produce recurrent modes equal to one, and that the input-output modes learn to scale these dynamics. In the case where the output is a sum of inputs, no additional scaling is necessary so input-output modes should become one, while in the case where the output is the mean, the input-output modes should become $1/T$ to appropriately scale.

Our theory also predicts that task dynamics that produce solutions dependent on trajectory length will be unable to extrapolate to other trajectory lengths. Thus, since input-output modes should learn $1/T$ for the mean-integration task, we do not expect it to extrapolate to other trajectory lengths, while we would expect the sum-integration task to extrapolate perfectly.

By simulating networks on the sensory integration tasks and plotting the network SVs, we see that our theory indeed predicts the solutions found by networks for both mean-integration and sum-integration (top row in Figure 6). As expected, we also find that the networks trained on sum-integration tasks are able to extrapolate to other trajectory lengths perfectly, while networks trained on mean-integration accumulate error as a function of the difference in trajectory length from that trained on (bottom row in Figure 6). In Appendix L, we further extend this setting to show that our predictions about stability (early-importance versus late-importance dynamics) are validated in networks without our assumptions. In summary, these results illustrate the application of our theoretical framework for understanding the behavior and capabilities of LRNNs more generally.

## 4. Discussion & Related Work

**Summary of results.** In this work, we extend the growing literature on learning dynamics to a new architecture, linear RNNs. We derive an analytical solution to the energy function and learning dynamics of LRNNs under certain conditions, using a novel approach that accounts for task dynamics. Unlike feedforward networks, LRNNs learn data singular values ordered by both their scale and temporal precedence, with larger and later singular values being learned first. We identify how task dynamics impact solution stability and extrapolation ability, an often understudied aspect of RNN dynamics. We further reveal a tradeoff between recurrent and feedforward computation that leads to low-rank solutions, mediated by an effective regularization term in the energy function. We extend existing work on rich and lazy learning in RNNs beyond the effect of initial connectivity by deriving the NTK for finite-width LRNNs and showing that recurrence encourages feature learning. Finally, we demonstrate an application of our results in a sensory integration task where we relax our prior assumptions and find that our theory explains the behavior of LRNNs.

**Learning dynamics in linear networks.** Differing from prior work on learning dynamics in linear networks (Saxe et al., 2014; 2018; 2022; Braun et al., 2022; Sandbrink et al., 2024; Dominé et al., 2025), we study a recurrent network, allowing us to analyze how other architectures constrain optimization in ways that differ from feedforward ones. Notably, Schuessler et al. (2020b) previously studied learning

dynamics in LRNNs to study how networks make low-rank changes to their connectivity, but used a task with constant input in the limit of infinite trajectory length, and Smékal et al. (2024) studied learning in the frequency domain but focused on the effects of overparameterization on convergence time. Instead, our work accounts for the effect of task dynamics, which are critically important for modeling and understanding dynamic cognitive behavior.

**Stability and extrapolation.** The problem of stability in training RNNs is a well-studied problem (Bengio et al., 1994; Hochreiter et al., 2001; Pascanu et al., 2012; Zucchet & Orvieto, 2024) with numerous proposed solutions (Hochreiter & Schmidhuber, 1997; Le et al., 2015; Orvieto et al., 2023; Zucchet et al., 2023). Here, we highlight an additional, understudied factor — the impact of task dynamics. We show how certain task dynamics (those with early-importance) can lead to unstable training regimes as a result of the solutions they drive the network to. This suggests that practical approaches to such problems should take task dynamics into account when designing new solutions. Although less theoretically understood (Emami et al., 2021; Cohen-Karlik et al., 2022; Beirán et al., 2023), our framework sheds light on how task dynamics impact LRNN's extrapolation to sequence lengths different to those in the training set and how this is driven by a mismatch between architecture and the latent structure of the data.

**Low-rank connectivity.** Networks with low-rank connectivity have been used as more interpretable models from which to study dynamics related to cognition (Mastrogiuseppe & Ostojic, 2018; Schuessler et al., 2020a;b; Dubreuil et al., 2022), motivated by the fact that neural population activity is often low-dimensional, and it's been shown that RNNs learn low-rank solutions along task dimensions (Schuessler et al., 2020b). Complementing this work, we identify an effective regularization term in the energy function that incentivizes small-weight solutions and demonstrate specific cases of task dynamics where RNNs prune connectivity modes resulting in low-rank connectivity.

**Rich and lazy learning.** Neural networks can lie in two different learning regimes (so-called rich or lazy) depending on their weight initialization and width, and there is increasing evidence that these regimes are related to the representational geometry of different brain regions (Rigotti et al., 2013; Bernardi et al., 2020; Flesch et al., 2022; Farrell et al., 2023; Payeur et al., 2023). Most theoretical studies of rich and lazy learning have been done in feedforward networks, with the exception of Liu et al. (2024), which showed that connectivity rank impacts features learning, and Schuessler et al. (2024), which showed that the scale of the readout acts as a control parameter between aligned and oblique dynamics in RNNs. Here we reveal an additional factor impacting feature learning: the effect of recurrence. Although lazy learning is still possible in recurrent networks (e.g., with larger weight initializations or widths), we find that recurrence induces substantial NTK movement.

**Limitations and future directions.** In this work, we perform our theoretical analysis on LRNNs with *data-aligned* weights, trained on tasks with input-output correlations that have constant singular/eigen-vectors. While our form based on SVD severely restricts the expressivity of the network, our derivation based on an eigendecomposition (Appendix N) relieves many of these limitations, whereby our framework and results naturally extend to this case. Interestingly, other work has shown that there are some practical justifications for using diagonal state spaces (Gupta & Berant, 2022; Orvieto et al., 2023), and it's also a common choice in theoretical work (Hazan et al., 2018; Zucchet & Orvieto, 2024).

In this paper, we make several choices when constructing the setting we study, including our focus on the single-output case (rather than an autoregressive one), our initialization of the hidden layer at 0, and our use of small square networks. While we do extend our main derivations to the autoregressive ($T$-output) case, fully characterizing the behavior of RNNs will require charting these different settings.

Finally, although linear networks are more tractable, many computations of interest can only be implemented in RNNs with nonlinear dynamics. Thus, an important future direction of theory will be to find new ways to study learning in networks with nonlinearities, potentially through gating (Saxe et al., 2022; Sandbrink et al., 2024; Jarvis et al., 2025). It's an open question to what extent the findings in this work will generalize to other settings, but we believe the framework we have constructed is flexible and will support new research inquiries in this direction.

## 5. Conclusion

This work presents a theoretical study of learning dynamics in linear RNNs and the effect of temporally-structured data. It presents one of the few studies of learning dynamics in recurrent networks, and, to our best knowledge, the first to account for the effect of task dynamics and to more explicitly connect recurrence to feature learning by studying the transition from feedforward to recurrent networks using trajectory length. This study generates new insights into the learning process of RNNs and encourages further theoretical developments to consider the learning process and the impact of temporal data when studying RNNs. We hope future work can characterize how complex dynamics, such as those in the brain, are developed during learning and ultimately, help us better understand cognition from a dynamic perspective.

## Acknowledgements

AP is funded by the Imperial College London President's PhD Scholarship. CD was supported by the Gatsby Charitable Foundation (GAT3755). This research was funded in part by the Wellcome Trust [216386/Z/19/Z].

## Impact Statement

This paper presents work whose goal is to advance the field of Machine Learning. There are many potential societal consequences of our work, none which we feel must be specifically highlighted here.

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

# Appendix

# Table of Contents

# A. Notation

*Table A1.* Notation

| Symbol | Description |
| --- | --- |
| $t$ | trajectory timestep |
| $T$ | trajectory length (final timestep) |
| $t_\theta$ | learning timestep |
| $\tau$ | learning timescale (inverse learning rate) |
| $\eta$ | learning rate |
| $P$ | dataset size |
| $p$ | data sample index |
| $N_x$ | input size |
| $N_h$ | hidden size |
| $N_y$ | output size |
| $\boldsymbol{x}_{p,t} \in \mathbb{R}^{N_x}$ | input sample $p$ at timestep $t$ |
| $\boldsymbol{h}_{t+1} = W_h \boldsymbol{h}_t + W_x \boldsymbol{x}_t \in \mathbb{R}^{N_h}$ | hidden state at timestep $t+1$ |
| $\hat{\boldsymbol{y}}_t = W_y \boldsymbol{h}_{t+1} \in \mathbb{R}^{N_y}$ | model output at timestep $t$ |
| $\boldsymbol{y}_{p,t} \in \mathbb{R}^{N_y}$ | output target at timestep $t$ |
| $W_x \in \mathbb{R}^{N_h \times N_x}$ | input weight matrix |
| $W_h \in \mathbb{R}^{N_h \times N_h}$ | recurrent weight matrix |
| $W_y \in \mathbb{R}^{N_y \times N_h}$ | output weight matrix |
| $\Sigma^{YX_t} = \sum_{p=1}^{P} \boldsymbol{y}_{p,T} \boldsymbol{x}_{p,t}^{\top}$ | input-output correlation matrix between input $\boldsymbol{x}_t$ at trajectory timestep $t$ and final output target $\boldsymbol{y}_T$ |
| $\Sigma^{X_t X_{t'}} = \sum_{p=1}^{P} \boldsymbol{x}_{p,t} \boldsymbol{x}_{p,t'}^{\top}$ | input-input correlation matrix between input $\boldsymbol{x}_t$ at trajectory timestep $t$ and $\boldsymbol{x}_{t'}$ at $t'$ |
| $U_y S_t V_x^{\top} = \Sigma^{YX_t}$ | singular value decomposition of input-output correlation matrix for input at timestep $t$ |
| $P D_t P^{\dagger} = \Sigma^{YX_t}$ | eigendecomposition of input-output correlation matrix for input at timestep $t$ |
| $\overline{W}_x$ | diagonalized input matrix |
| $\overline{W}_h$ | diagonalized recurrent matrix |
| $\overline{W}_y$ | diagonalized output matrix |
| $\alpha$ | singular/eigen-value dimension |
| $a_\alpha$ | input connectivity mode at dimension $\alpha$ |
| $b_\alpha$ | recurrent connectivity mode at dimension $\alpha$ |
| $c_\alpha$ | output connectivity mode at dimension $\alpha$ |
| $s_{\alpha,t}$ | singular value of $S_t$ at dimension $\alpha$ |
| $d_{\alpha,t}$ | eigenvalue of $D_t$ at dimension $\alpha$ |
| $E$ | energy function of connectivity modes decoupled along singular/eigen-value dimensions |
| $\delta$ | constant component/parameter of data singular/eigen-values (when $s_t, d_t = \delta f(\lambda, t)$) |
| $\lambda$ | dynamic component/parameter of data singular/eigen-values (when $s_t, d_t = \delta f(\lambda, t)$) |
| $f(\lambda, t) = 1$ | constant task dynamics |
| $f(\lambda, t) = \lambda^{T-t}$ | inverse-exponential task dynamics |
| $f(\lambda, t) = \lambda^t$ | exponential task dynamics |
| $\kappa$ | 'feedforward computation' ($= s_T$) in phase transition experiments |
| $\beta$ | 'recurrent computation' ($= s_1$) in Dirac delta task dynamics |
| $\tilde{\ }$ | indicating teacher parameters |
| $u = ac$ | single variable for input-output modes |
| $\star$ | indicating global solution |
| $\Sigma^{Y_t X_{t'}} = \sum_{p=1}^{P} \boldsymbol{y}_{p,t} \boldsymbol{x}_{p,t'}^{\top}$ | input-output correlation matrix between input at timestep $t'$ and output at timestep $t$ ($t' \leq t$) for autoregressive case |
| $\dagger$ | conjugate transpose of a matrix |
| $*$ | complex conjugate |
| $R_\delta$ | radial component of $\delta$ ($= R_\delta e^{\phi_\delta i}$) |
| $\phi_\delta$ | angle component of $\delta$ |
| $R_\lambda$ | radial component of $\lambda$ ($= R_\lambda e^{\phi_\lambda i}$) |
| $\phi_\lambda$ | angle component of $\lambda$ |

# B. Derivation of gradient flow equations and energy function

Recall our model definition as

$$\boldsymbol{h}_{t+1} = W_h \boldsymbol{h}_t + W_x \boldsymbol{x}_t \tag{12}$$

$$= \sum_{i=1}^{t} W_h^{t-i} W_x \boldsymbol{x}_i \tag{13}$$

$$\hat{\boldsymbol{y}}_t = W_y \boldsymbol{h}_{t+1} \tag{14}$$

with a loss of

$$\mathcal{L} = \frac{1}{2} \sum_{p=1}^{P} \| \boldsymbol{y}_{p,T} - W_y \big( \sum_{i=1}^{T} W_h^{T-i} W_x \boldsymbol{x}_{p,i} \big) \|^2 \tag{15}$$

By taking the derivative of the loss with respect to each set of parameters $W_x, W_h, W_y$, we get the following equations

$$\frac{\partial \mathcal{L}}{\partial W_x} = - \sum_{p=1}^{P} \left[ \sum_{i=1}^{T} \left( W_h^{(T-i)\top} W_y^{\top} \left( \boldsymbol{y}_{T,p} - \sum_{j=1}^{T} W_y W_h^{T-j} W_x \boldsymbol{x}_{j,p} \right) \boldsymbol{x}_{i,p}^{\top} \right) \right] \tag{16}$$

$$\frac{\partial \mathcal{L}}{\partial W_h} = - \sum_{p=1}^{P} \left[ \sum_{i=1}^{T-1} \sum_{r=0}^{T-i-1} \left( W_h^{(r)\top} W_y^{\top} \left( \boldsymbol{y}_{T,p} - \sum_{j=1}^{T} W_y W_h^{T-j} W_x \boldsymbol{x}_{j,p} \right) \boldsymbol{x}_{i,p}^{\top} W_x^{\top} W_h^{(T-i-1-r)\top} \right) \right] \tag{17}$$

$$\frac{\partial \mathcal{L}}{\partial W_y} = - \sum_{p=1}^{P} \left[ \sum_{i=1}^{T} \left( \left( \boldsymbol{y}_{T,p} - \sum_{j=1}^{T} W_y W_h^{T-j} W_x \boldsymbol{x}_{j,p} \right) \boldsymbol{x}_{i,p}^{\top} W_x^{\top} W_h^{(T-i)\top} \right) \right] \tag{18}$$

We define the input-output correlation matrices between an input at trajectory timestep $t$ and the final output as

$$\Sigma^{YX_t} = \sum_{p=1}^{P} \boldsymbol{y}_{p,T} \boldsymbol{x}_{p,t}^{\top} \tag{19}$$

$$\tag{20}$$

and the input-input correlation matrices between two inputs at trajectory timesteps $t, t'$ as

$$\Sigma^{X_t X_{t'}} = \sum_{p=1}^{P} \boldsymbol{x}_{p,t} \boldsymbol{x}_{p,t'}^{\top} \tag{21}$$

$$\tag{22}$$

Under the assumption of whitened input with 0 mean, the input-input correlation matrices become $\Sigma^{X_t X_{t'}} = \mathbf{0}, \forall t \neq t'$ and $\Sigma^{X_t X_t} = I$. Substituting the correlation matrices and assuming the gradient flow regime where the learning rate ($\eta = 1/\tau$) is small, we can rewrite the gradient equations above as a set of differential equations over training time $t_\theta$

$$\tau \frac{d}{dt_\theta} W_x = \sum_{i=1}^{T} W_h^{(T-i)\top} W_y^{\top} (\Sigma^{YX_i} - W_y W_h^{T-i} W_x) \tag{23}$$

$$\tau \frac{d}{dt_\theta} W_h = \sum_{i=1}^{T-1} \sum_{r=0}^{T-i-1} W_h^{(r)\top} W_y^{\top} (\Sigma^{YX_i} - W_y W_h^{T-i} W_x) W_x^{\top} W_h^{(T-i-1-r)\top} \tag{24}$$

$$\tau \frac{d}{dt_\theta} W_y = \sum_{i=1}^{T} (\Sigma^{YX_i} - W_y W_h^{T-i} W_x) W_x^{\top} W_h^{(T-i)\top} \tag{25}$$

We assume that the input-output correlation matrices have constant left and right singular vectors across trajectory timesteps, such that only their singular values vary through time. Although this may seem like a restrictive assumption, note that this assumption holds for any data generated by a teacher linear RNN with weights that can be diagonalized.

*Proof.* Data generated by a linear RNN teacher parameterized by $\tilde{W}_x, \tilde{W}_h, \tilde{W}_y$ that can be diagonalized with SVD has constant left and right singular vectors across trajectory timesteps.

$$\Sigma^{YX_t} = \sum_{p=1}^{P} \boldsymbol{y}_{p,T} \boldsymbol{x}_{p,t}^{\top} \tag{26}$$

$$= \sum_{p=1}^{P} \tilde{W}_y \tilde{W}_h^{T-t} \tilde{W}_x \boldsymbol{x}_{p,t} \boldsymbol{x}_{p,t}^{\top} \tag{27}$$

$$= \tilde{W}_y \tilde{W}_h^{T-t} \tilde{W}_x \tag{28}$$

We assume $\tilde{W}_h$ is diagonalized by orthogonal matrices $\tilde{R}_y, \tilde{R}_x$ such that $\tilde{W}_y = U_y S_y \tilde{R}_y^{\top}, \tilde{W}_x = \tilde{R}_x S_x V_x^{\top}, \tilde{W}_h = \tilde{R}_y S_h \tilde{R}_x^{\top}$. Then,

$$\tilde{W}_y \tilde{W}_h^{T-t} \tilde{W}_x = U_y S_t V_x^{\top} \tag{29}$$

$\square$

We place no additional assumptions on the temporal dynamics of the singular values through time $S_t$, such that they could be generated by any dynamic process. Substituting the singular value decomposition (SVD) of the data-correlation matrix into the gradient flow equations yields

$$\tau \frac{d}{dt_\theta} W_x = \sum_{i=1}^{T} W_h^{(T-i)\top} W_y^{\top} (U_y S_i V_x^{\top} - W_y W_h^{T-i} W_x) \tag{30}$$

$$\tau \frac{d}{dt_\theta} W_h = \sum_{i=1}^{T-1} \sum_{r=0}^{T-i-1} W_h^{(r)\top} W_y^{\top} (U_y S_i V_x^{\top} - W_y W_h^{T-i} W_x) W_x^{\top} W_h^{(T-i-1-r)\top} \tag{31}$$

$$\tau \frac{d}{dt_\theta} W_y = \sum_{i=1}^{T} (U_y S_i V_x^{\top} - W_y W_h^{T-i} W_x) W_x^{\top} W_h^{(T-i)\top} \tag{32}$$

Similarly to Saxe et al. (2014; 2018), we assume the LRNN is *data-aligned* at initialization such that for some orthogonal matrices $R_y, R_x$, $R_y^{\top} W_h(0) R_x = \overline{W}_h(0)$, $R_x^{\top} W_x(0) V_x = \overline{W}_x(0)$, $U_y^{\top} W_y(0) R_y = \overline{W}_y(0)$, where $\overline{W}_x, \overline{W}_h, \overline{W}_y$ are diagonal matrices. Atanasov et al. (2022) showed that this alignment happens early in training for networks initialized with small random weights. Performing a change of variables in the gradient flow equations and simplifying yields,

$$\tau \frac{d}{dt_\theta} \overline{W}_x = \sum_{i=1}^{T} \overline{W}_h^{(T-i)\top} \overline{W}_y^{\top} (S_i - \overline{W}_y \overline{W}_h^{T-i} \overline{W}_x) \tag{33}$$

$$\tau \frac{d}{dt_\theta} \overline{W}_h = \sum_{i=1}^{T-1} \sum_{r=0}^{T-i-1} \overline{W}_h^{(r)\top} \overline{W}_y^{\top} (S_i - \overline{W}_y \overline{W}_h^{T-i} \overline{W}_x) \overline{W}_x^{\top} \overline{W}_h^{(T-i-1-r)\top} \tag{34}$$

$$\tau \frac{d}{dt_\theta} \overline{W}_y = \sum_{i=1}^{T} (S_i - \overline{W}_y \overline{W}_h^{T-i} \overline{W}_x) \overline{W}_x^{\top} \overline{W}_h^{(T-i)\top} \tag{35}$$

Let $a_\alpha, b_\alpha, c_\alpha$ be the $\alpha^{\text{th}}$ diagonal entry of $\overline{W}_x, \overline{W}_h, \overline{W}_y$, respectively, and $s_{\alpha,t}$ be the $\alpha^{\text{th}}$ singular value of $S_t$. We can then rewrite the above equations in terms of these variables, or *connectivity modes* that decouple along singular value dimensions

$\alpha$,

$$\tau\frac{d}{dt_\theta}a_\alpha = \sum_{i=1}^{T}b_\alpha^{T-i}c_\alpha(s_{\alpha,i} - c_\alpha b_\alpha^{T-i}a_\alpha) \tag{36}$$

$$\tau\frac{d}{dt_\theta}b_\alpha = \sum_{i=1}^{T-1}\sum_{r=0}^{T-i-1}b_\alpha^{(r)}c_\alpha(s_{\alpha,i} - c_\alpha b_\alpha^{T-i}a_\alpha)a_\alpha b_\alpha^{(T-i-1-r)} \tag{37}$$

$$= \sum_{i=1}^{T-1}(T-i)c_\alpha(s_{\alpha,i} - c_\alpha b_\alpha^{T-i}a_\alpha)a_\alpha b_\alpha^{(T-i-1)} \tag{38}$$

$$\tau\frac{d}{dt_\theta}c_\alpha = \sum_{i=1}^{T}(s_{\alpha,i} - c_\alpha b_\alpha^{T-i}a_\alpha)a_\alpha b_\alpha^{T-i} \tag{39}$$

These dynamics arise from gradient descent on the energy function

$$E = \frac{1}{2\tau}\sum_\alpha\sum_{i=1}^{T}(s_{\alpha,i} - c_\alpha b_\alpha^{T-i}a_\alpha)^2 \tag{40}$$

To ease notation, we omit specifying $\alpha$ when referring to connectivity modes, although note that all terms $(s_t, a, b, c)$ still refer to a particular singular value dimension $\alpha$.

## C. Exact solution of input-output connectivity modes

We solve for the learning dynamics of the input-output connectivity modes when the recurrent connectivity mode is frozen. If we assume balanced weights such that $a = c$, we can solve for both modes $u = ac$ together.

$$\tau\frac{d}{dt_\theta}u = c(\tau\frac{d}{dt_\theta}a) + a(\tau\frac{d}{dt_\theta}c) \tag{41}$$

$$= c(\sum_{i=1}^{T}b^{T-i}c(s_i - cb^{T-i}a)) + a(\sum_{i=1}^{T}ab^{T-i}(s - cb^{T-i}a)) \tag{42}$$

$$= 2u(\sum_{i=1}^{T}b^{T-i}(s_i - b^{T-i}u)) \tag{43}$$

This equation can be integrated to yield

$$t_\theta = \tau\int_{u(0)}^{u(t_\theta)}\frac{du}{\sum_{i=1}^{T}2ub^{T-i}(s_i - ub^{T-i})} \tag{44}$$

$$= \frac{\tau}{2}\frac{\log(u) - \log(\sum_{i=1}^{T}b^{T-i}s_i - ub^{2(T-i)})}{\sum_{i=1}^{T}b^{T-i}s_i}\Big|_{u(0)}^{u(t_\theta)} \tag{45}$$

$$= \frac{\tau}{2\sum_{i=1}^{T}b^{T-i}s_i}\log\frac{u(t_\theta)(\sum_{i=1}^{T}b^{T-i}s_i - u(0)b^{2(T-i)})}{u(0)(\sum_{i=1}^{T}b^{T-i}s_i - u(t_\theta)b^{2(T-i)})} \tag{46}$$

$$u(t_\theta) = \frac{e^{2t_\theta(\sum_{i=1}^{T}b^{T-i}s_i)/\tau}(\sum_{i=1}^{T}b^{T-i}s_i)}{(\sum_{i=1}^{T}b^{T-i}s_i)/u(0) - (\sum_{i=1}^{T}b^{2(T-i)}) + e^{2t_\theta(\sum_{i=1}^{T}b^{T-i}s_i)/\tau}(\sum_{i=1}^{T}b^{2(T-i)})} \tag{47}$$

## D. Local approximation of recurrent connectivity modes

Due to the exponential term, the learning dynamics of the recurrent mode $b$ are difficult to solve for. Instead, we take an approach similar to Schuessler et al. (2020b), by performing a Taylor expansion on the learning dynamics of $b$ through training time (with input-output modes held constant),

$$b(t_\theta/\tau) = \sum_{n=0}^{\infty}\frac{d^n b(0)}{dt_\theta^n}\frac{(t_\theta/\tau)^n}{n!} \tag{48}$$

First we solve for the $n$th partial derivative of the energy function $E$ with respect to the recurrent mode $b$ which has an explicit closed-form solution given by

$$\frac{d^n E}{db^n} = \sum_{i=1}^{T-n} \left[ (T-i) \left( \prod_{j=1}^{n-1} (T-i-j) \right) s_i c a b^{T-i-n} \right] - \sum_{i=1}^{(2T-n)/2} \left[ (T-i) \left( \prod_{j=1}^{n-1} (2T-2i-j) \right) c^2 a^2 b^{2T-2i-n} \right]$$
(49)

Recall that in the gradient flow regime, the recurrent mode $b$ changes continuously according to $\frac{db}{dt_\theta} = \frac{dE}{db}$. Thus, to compute higher-order derivatives of $b$,

$$\frac{d^n b}{dt_\theta^n} = \frac{d}{dt_\theta} \left( \frac{d^{n-1} b}{dt_\theta^{n-1}} \right)$$
(50)

$$= \frac{d}{db} \left( \frac{d^{n-1} b}{dt_\theta^{n-1}} \right) \frac{db}{dt_\theta}$$
(51)

$$= \frac{d}{db} \left( \frac{d^{n-1} b}{dt_\theta^{n-1}} \right) \frac{dE}{db}.$$
(52)

Note that this is a recursive operation and does not give a simple closed-form expression.

Applying this to higher-orders and using chain rule, we compute the time-derivatives of $b$ up to 5th order,

$$\frac{db}{dt_\theta} = \frac{dE}{db}$$
(53)

$$\frac{d^2 b}{dt_\theta^2} = \frac{d^2 E}{db^2} \frac{dE}{db}$$
(54)

$$\frac{d^3 b}{dt_\theta^3} = \left( (\frac{d^2 E}{db^2})^2 + \frac{d^3 E}{db^3} \frac{dE}{db} \right) \frac{dE}{db}$$
(55)

$$\frac{d^4 b}{dt_\theta^4} = \left( 4 \frac{d^2 E}{db^2} \frac{d^3 E}{db^3} (\frac{dE}{db})^1 + (\frac{d^2 E}{db^2})^3 + \frac{d^4 E}{db^4} (\frac{dE}{db})^2 \right) \frac{dE}{db}$$
(56)

$$\frac{d^5 b}{dt^5} = \left( (\frac{d^2 E}{db^2})^4 + 11 (\frac{d^2 E}{db^2})^2 \frac{d^3 E}{db^3} (\frac{dE}{db}) + 4 (\frac{d^3 E}{db^3})^2 (\frac{dE}{db})^2 + 7 \frac{d^4 E}{db^4} \frac{d^2 E}{db^2} (\frac{dE}{db})^2 + \frac{d^5 E}{db^5} (\frac{dE}{db})^3 \right) \frac{dE}{db}$$
(57)

We then approximate the learning dynamics of the recurrent mode $b$ and substitute the formula above for the $n$th partial derivative of the energy function

$$b(t_\theta/\tau) \approx b(0) + \frac{dE}{db} (t_\theta/\tau) + \frac{d^2 E}{db^2} \frac{dE}{db} \frac{(t_\theta/\tau)^2}{2!} + \left( (\frac{d^2 E}{db^2})^2 + \frac{d^3 E}{db^3} \frac{dE}{db} \right) \frac{dE}{db} \frac{(t_\theta/\tau)^3}{3!}$$
(58)

$$+ \left( 4 \frac{d^2 E}{db^2} \frac{d^3 E}{db^3} (\frac{dE}{db})^1 + (\frac{d^2 E}{db^2})^3 + \frac{d^4 E}{db^4} (\frac{dE}{db})^2 \right) \frac{dE}{db} \frac{(t_\theta/\tau)^4}{4!}$$
(59)

$$+ \left( (\frac{d^2 E}{db^2})^4 + 11 (\frac{d^2 E}{db^2})^2 \frac{d^3 E}{db^3} (\frac{dE}{db}) + 4 (\frac{d^3 E}{db^3})^2 (\frac{dE}{db})^2 + 7 \frac{d^4 E}{db^4} \frac{d^2 E}{db^2} (\frac{dE}{db})^2 + \frac{d^5 E}{db^5} (\frac{dE}{db})^3 \right) \frac{dE}{db} \frac{(t_\theta/\tau)^5}{5!}$$
(60)

$$+ O(6)$$
(61)

In practice, when simulating the learning dynamics using this approximation, we apply the solution locally across a window of size $\Delta$ and iterate over each window $b(t_\theta : t_\theta + \Delta)$. The window-size is dependent on the smoothness of the connectivity mode dynamics (i.e., how sharp the gradient is).

### D.1. Analytical approximation using Faà di Bruno formula and Bell polynomials

Here we use the Faà di Bruno formula/Bell polynomials to write out a combinatorial solution for the $n$th derivative of $b$, which can be used to expand the learning dynamics of $b$ to higher orders without repeated recursive chain rule. Using this

approach, the $n$th derivative of $b$ is

$$\frac{d^n b}{dt_\theta^n} = \sum_{k=1}^{n-1} \frac{d^{k+1} E}{db^{k+1}} B_{n-1,k}\left[\frac{db}{dt_\theta}, \frac{d^2 b}{dt_\theta^2}, \ldots, \frac{d^{n-k} b}{dt_\theta^{n-k}}\right] \tag{62}$$

$$= \sum_{k=1}^{n-1} \frac{d^{k+1} E}{db^{k+1}} \sum_{\{m_1, m_2, \ldots, m_{n-k}\}} \frac{(n-1)!}{m_1! m_2! \ldots m_{n-k}!} \prod_{j=1}^{n-k} \frac{1}{j!^{m_j}} \left(\frac{d^j b}{dt_\theta^j}\right)^{m_j} \tag{63}$$

where the summation over $\{m_1, m_2, \ldots, m_{n-k}\}$ indicates a summation over all $n - k$ partitions of nonnegative integers satisfying

$$m_1 + m_2 + \cdots + m_{n-k} = k \tag{64}$$

$$1 m_1 + 2 m_2 + \cdots + (n-k) m_{n-k} = n_1 \tag{65}$$

Although useful, we note that this approach still requires substitution of other lower-order terms of $b$ (because of the $\frac{d^j b}{dt_\theta^j}$ term). The equation can be substituted back into the Taylor expansion of $b$ through training time,

$$b(t_\theta/\tau) = \sum_{n=0}^{\infty} \frac{d^n b(0)}{dt_\theta^n} \frac{(t_\theta/\tau)^n}{n!} \tag{66}$$

$$= b(0) + \frac{dE}{db}(t_\theta/\tau) + \sum_{n=2}^{\infty} \left(\sum_{k=1}^{n-1} \frac{d^{k+1} E}{db^{k+1}} \sum_{\{m_1, m_2, \ldots, m_{n-k}\}} \frac{(n-1)!}{m_1! m_2! \ldots m_{n-k}!} \prod_{j=1}^{n-k} \frac{1}{j!^{m_j}} \left(\frac{d^j b}{dt_\theta^j}\right)^{m_j}\right) \frac{(t_\theta/\tau)^n}{n!} \tag{67}$$

## E. Effect of task dynamics on the ordering of learning

To illustrate the effect of the ordering of singular values on learning speed, we compare task dynamics where the network connectivity modes learn solutions of the same magnitude (for different modes such that input-output and recurrent modes "swap" solutions), but the ordering of singular values is either ascending or descending. More specifically, we consider the case of inverse-exponential task dynamics given by $s_t = \delta f(\lambda, t); f(\lambda, t) = \lambda^{T-t}$. The solution for inverse-exponential dynamics are $ac = \delta, b = \lambda$. Thus, we switch the values for $\delta, \lambda$ (orange: $\delta = 1.1, \lambda = 0.5$, blue: $\delta = 0.5, \lambda = 1.1$) for two simulations so that the network connectivity modes learn solutions of the same magnitude (so that we somewhat control for the effect that larger singular values has on accelerating learning speed), but the task dynamics either have ascending or descending singular values over the trajectory length, and, more importantly, the magnitude of the SVs at the end of the trajectory differ. We indeed see that modes trained on task dynamics with larger singular values occurring later in the trajectory learn faster. As we discuss in the main text, this is due to the fact that the recurrent connectivity mode $b$ scales the gradient contribution for early trajectory timesteps exponentially. Since $b$ is initialized to be less than 1, this has the effect of downscaling the gradient contribution of earlier timesteps compared to later ones. This manifests in singular values occurring later in the trajectory to "contribute more" to learning (when $b < 1$), such that modes trained on task dynamics with larger and later singular values learn faster (because they have larger gradient updates).

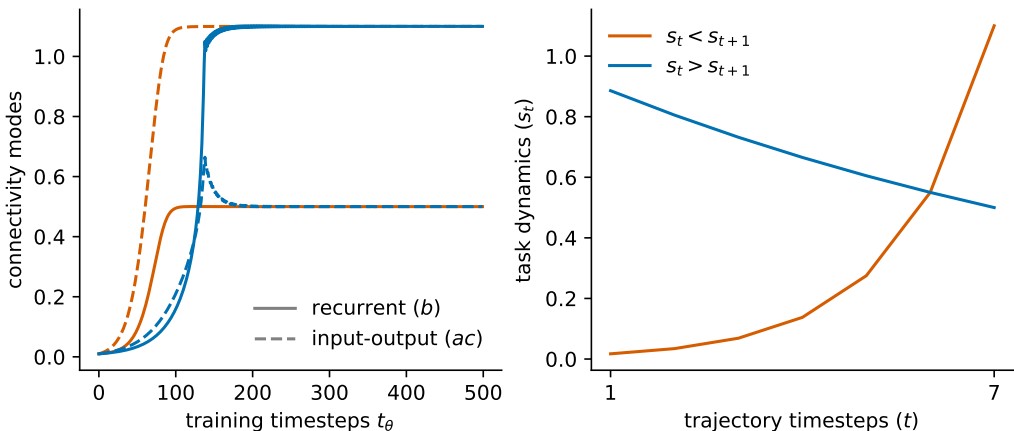

*Figure A1.* **Large singular values occurring later in trajectory of the task dynamics accelerate learning compared to earlier ones.** Networks trained on task dynamics with larger singular values occurring *later* in the trajectory learn faster.

This relates to the well-studied problems of learning of long-range dependencies and vanishing gradients in RNNs, as data from earlier timesteps are harder to learn as trajectory length increases because of the exponential downscaling effect of the recurrent mode. Here, we build on these ideas to understand learning speed and its dependency on task dynamics, including both the ordering of singular values and their scale.

This also relates to our study on the tradeoff between feedforward versus recurrent computation in LRNNs in Section 3.4. Because later singular values have a greater effect on the gradient of the loss, the feedforward computation is effectively favored in learning, although the cumulative effect of the recurrent computation can of course outweigh this.

## F. Zero-loss solutions only exist for inverse-exponential task dynamics

### F.1. Discussion on exponential task dynamics as reparameterization of inverse-exponential task dynamics

In the main text, we refer to inverse-exponential and exponential task dynamics separately. Here, we make the distinction that although we refer to these separately, they can both be rewritten as reparameterizations of each other (i.e., exponential task dynamics can be rewritten in the inverse-exponential form and vice versa) specifically when the trajectory length is held constant. We refer to inverse-exponential and exponential dynamics separately in the main text primarily to distinguish between cases that extrapolate (or don't) and illustrate how RNNs can learn perfect solutions that do not match the ground-truth data generating process.

In particular, for fixed $T$, exponential task dynamics given by $s_t = \delta f(\lambda, t)$ where $f(\lambda, t) = \lambda^t$ can equivalently be written as $s_t = \delta \lambda^T g(\lambda, t)$ for $g(\lambda, t) = \left(\frac{1}{\lambda}\right)^{T-t}$.

This illustrates how and why LRNNs can still learn a perfect solution to exponential task dynamics for a fixed trajectory length (by overfitting), but because their architecture does not match the latent structure of the ground-truth data generating process, the network will not extrapolate to other trajectory lengths. More generally, we can see how mismatches between latent task dynamics and the network's recurrent dynamics can lead to non-extrapolating solutions. Of course, this does not occur when data is generated by a teacher network with a matched architecture because the latent form of the task dynamics and recurrent dynamics are the same.

### F.2. Proof

Here we prove that zero-loss solutions only exist for inverse-exponential task dynamics. Recall the energy function is given by

$$E = \frac{1}{2\tau} \sum_{i=1}^{T} (s_i - cb^{T-i}a)^2 \tag{68}$$

The only task dynamics with zero-loss solutions in aligned LRNNs are those with inverse-exponential task dynamics $(s_t = \delta\lambda^{T-t} \, \forall t)$.

*Proof.* The term $(s_i - cb^{T-i}a)^2$ is a quadratic function and thus is nonnegative for real numbers, meaning that the 0-loss solution to the energy function must satisfy $(s_i - cb^{T-i}a) = 0 \, \forall i$. Furthermore, because $(s_i - cb^{T-i}a)^2$ is a quadratic function, there exists a unique global minimum when $s_i = cb^{T-i}a$. Thus, the only zero-loss solutions exist at $cb^{T-i}a = s_i$, which can only be all satisfied simultaneously when $s_i = \delta\lambda^{T-i}$. $\qquad\square$

### F.3. Global solutions of task dynamics

Here, we solve for the global solutions of constant, inverse-exponential, and exponential task dynamics.

#### F.3.1. CONSTANT TASK DYNAMICS

Constant task dynamics are defined as singular values $s_t = \delta f(\lambda, t)$ where $f(\lambda, t) = 1, \forall t$, such that $s_{1:T} = \delta$. The global solution for constant task dynamics is given by $b = 1, ac = \delta$.

*Proof.* The 0-loss solution corresponding to the global minimum given by $ac, b$ must satisfy $\sum_{i=1}^{T}(s_i - cb^{T-i}a) = 0$. This condition will be satisfied if $cb^{T-i}a = s_i \, \forall i$. For singular values following constant dynamics, this becomes

$$\sum_{i=1}^{T}(\delta - cb^{T-i}a) \tag{69}$$

At $i = T$, the expression becomes

$$cb^{T-T}a = \delta \tag{70}$$
$$ca = \delta \tag{71}$$

Then, substituting $ca = \delta$ when $i = T - 1$,

$$cb^{T-(T-1)}a = \delta \tag{72}$$
$$cba = \delta \tag{73}$$
$$b = 1 \tag{74}$$

Substituting the solution into our original expression

$$\sum_{i=1}^{T}(\delta - \delta(1)^{T-i}) \tag{75}$$

$$\sum_{i=1}^{T}(\delta - \delta) = 0 \tag{76}$$

$$\square$$

#### F.3.2. INVERSE-EXPONENTIAL TASK DYNAMICS

Inverse-exponential task dynamics are defined as singular values $s_t = \delta f(\lambda, t)$ where $f(\lambda, t) = \lambda^{T-t}$. The global solution for inverse-exponential task dynamics is given by $b = \lambda, ac = \delta$.

*Proof.* The 0-loss solution corresponding to the global minimum given by $ac, b$ must satisfy $\sum_{i=1}^{T}(s_i - cb^{T-i}a) = 0$. This condition will be satisfied if $cb^{T-i}a = s_i \, \forall i$. For singular values following inverse-exponential dynamics, this becomes

$$\sum_{i=1}^{T}(\delta\lambda^{T-i} - cb^{T-i}a) \tag{77}$$

At $i = T$, the expression becomes

$$cb^{T-T}a = \delta\lambda^{T-T} \tag{78}$$

$$ca = \delta \tag{79}$$

Then, substituting $ca = \delta$ when $i = T - 1$,

$$cb^{T-(T-1)}a = \delta\lambda^{T-(T-1)} \tag{80}$$

$$cba = \delta\lambda \tag{81}$$

$$b = \lambda \tag{82}$$

Substituting the solution into our original expression

$$\sum_{i=1}^{T}(\delta\lambda^{T-i} - \delta\lambda^{T-i}) = 0 \tag{83}$$

$\square$

### F.3.3. EXPONENTIAL TASK DYNAMICS

Exponential task dynamics are defined as singular values $s_t = \delta f(\lambda, t)$ where $f(\lambda, t) = \lambda^t$. The global solution for exponential task dynamics is given by $b = 1/\lambda, ac = \delta\lambda^T$.

*Proof.* The 0-loss solution corresponding to the global minimum given by $ac, b$ must satisfy $\sum_{i=1}^{T}(s_i - cb^{T-i}a) = 0$. This condition will be satisfied if $cb^{T-i}a = s_i \ \forall i$. For singular values following exponential dynamics, this becomes

$$\sum_{i=1}^{T}(\delta\lambda^t - cb^{T-i}a) \tag{84}$$

At $i = T$, the expression becomes

$$cb^{T-T}a = \delta\lambda^T \tag{85}$$

$$ca = \delta\lambda^T \tag{86}$$

Then, substituting $ca = \delta\lambda^T$ when $i = T - 1$,

$$cb^{T-(T-1)}a = \delta\lambda^{T-1} \tag{87}$$

$$cba = \delta\frac{\lambda^T}{\lambda} \tag{88}$$

$$b = \frac{1}{\lambda} \tag{89}$$

Substituting the solution into our original expression

$$\sum_{i=1}^{T}(\delta\lambda^i - \delta\lambda^T(1/\lambda)^{T-i}) \tag{90}$$

$$\sum_{i=1}^{T}(\delta\lambda^i - \delta\lambda^T\lambda^{i-T}) \tag{91}$$

$$\sum_{i=1}^{T}(\delta\lambda^i - \delta\lambda^i) = 0 \tag{92}$$

$$\tag{93}$$

$\square$

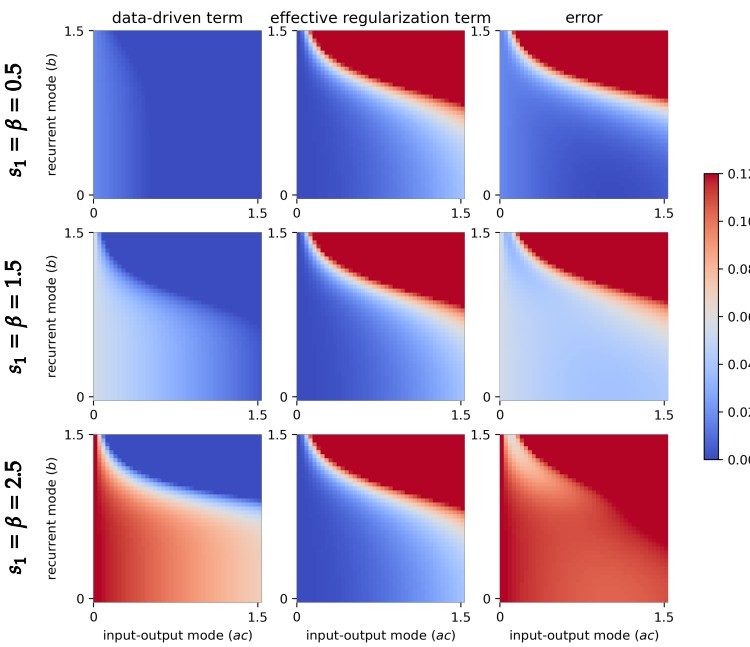

*Figure A2.* **Effective regularization term is constant across task dynamics and incentivizes small weights.** The phase plots show the energy function terms: (*left*) the data-driven term, (*middle*) the effective regularization, and (*right*) the error/energy, as a function of the connectivity modes for Dirac delta task dynamics with (*top*) $s_1 = 0.5$, (*middle*) $s_1 = 1.5$, and (*bottom*) $s_1 = 2.5$.

.

## G. Effective regularization term incentivizes small-weights

As discussed in Section 3.4, the energy function can be split into a data-driven term and an effective regularization term. Note that the form of the effective regularization term comes from the fact that $\sum_{i=1}^{T} b^{2(T-i)}$ is a geometric series that can be rewritten as $\frac{1-b^{2T}}{1-b^2}$. We plot the two terms, as well as their sum (the error), across different connectivity mode values to visualize the energy landscape (Figure A2). We do this for Dirac delta task dynamics where we keep the singular value at the last timestep constant ($s_T = 1$) and change the singular value at the first timestep across each row ($s_1 = \beta$). We can see how the effective regularization term affects the energy landscaping, forcing parameters to remain in a small-weight regime and how it mediates between the transition from feedforward computation to recurrent computation by forcing the input-output mode to decrease in magnitude when the recurrent mode increases in magnitude.

## H. Connectivity modes exhibit phase transition as a function of task dynamics

Here, we use Landau theory to show that the qualitative behavior we observe in Section 3.4 is characterized by a phase transition in the connectivity modes as a function of the ratio between recurrent and feedforward computations. We prove that for trajectory length $T = 3$, there is a smooth second-order phase transition and that for larger trajectory lengths $T > 3$, there are first-order phase transitions.

To simplify the problem, we consider Dirac delta task dynamics and show that the ratio of $s_1/s_T$ acts as a control parameter mediating the phase transition of connectivity modes up to a scaling.

*Proof.* The phase transition of the Dirac delta task dynamics is solely controlled by the ratio $s_1/s_T$ (up to a scaling). Treating the input-output weights as one term $u = ac$, the energy function for Dirac delta task dynamics can be written out as

$$E = (s_1 - ub^{T-1})^2 + (s_T - u)^2 + \sum_{i=2}^{T-1}(-ub^{T-i})^2 \tag{94}$$

Let $u' = u/s_T$. Then, by dividing the energy function by $s_T^2$, we get

$$\frac{E}{s_T^2} = (\frac{s_1}{s_T} - u'b^{T-1})^2 + (1 - u')^2 + \sum_{i=2}^{T-1}(-u'b^{T-i})^2 \tag{95}$$

We can see from this form that the impact of the data on the energy function can be reduced to a ratio of $s_1/s_T$, up to a scaling. $\square$

Because only the ratio matters, we can make the substitution $s_1/s_T = \beta^2$ to reduce the number of parameters. This is equivalent to the Dirac delta task dynamics given by

$$s_t = \begin{cases} \beta & \text{if } t = 1 \\ 1/\beta & \text{if } t = T \\ 0 & \text{otherwise} \end{cases} \tag{96}$$

In this form, as $\beta$ increases, the recurrent computation increases and the feedforward computation decreases. This simplification still captures the phenomena we're interested in: how the ratio between the recurrent and the feedforward computation affects the connectivity mode phase transition.

Returning back to the original form of the energy function,

$$E = (\beta - ub^{T-1})^2 + (\frac{1}{\beta} - u)^2 + \sum_{i=2}^{T-1}(-ub^{T-i})^2 \tag{97}$$

$$= \beta^2 - 2\beta ub^{T-1} + u^2 b^{2(T-1)} + \frac{1}{\beta^2} - \frac{2u}{\beta} + u^2 + u^2 \sum_{i=2}^{T-1} b^{2(T-i)} \tag{98}$$

$$= \beta^2 + \frac{1}{\beta^2} - 2u(\beta b^{T-1} + \frac{1}{\beta}) + u^2 \underbrace{\left(1 + b^{2(T-1)} + \sum_{i=2}^{T-1} b^{2(T-i)}\right)}_{\sum_{i=0}^{T-1} b^{2i}} \tag{99}$$

We can then solve for the minimum of the energy function with respect to the input-output modes $u$ (letting $u^\star$ denote the optimum)

$$\frac{\partial E}{\partial u} = -2(\beta b^{T-1} + \frac{1}{\beta}) + 2u^\star \sum_{i=0}^{T-1} b^{2i} = 0 \tag{100}$$

$$u^\star = \frac{\beta b^{T-1} + \frac{1}{\beta}}{\sum_{i=0}^{T-1} b^{2i}} \tag{101}$$

Substituting back into the energy function and simplifying yields the effective energy $E^\star(\beta)$ where $u$ is always at its minimum:

$$E^\star(\beta) = \beta^2 + \frac{1}{\beta^2} - \frac{(\beta b^{T-1} + \frac{1}{\beta})^2}{\sum_{i=0}^{T-1} b^{2i}} \tag{102}$$

The trivial (symmetric) solution to minimize $E$ is given by $b = 0, u = \frac{1}{\beta}$, yielding $E = \beta^2$. A phase transition requires that for some range of $\beta$, there is another solution where $b \neq 0$ and $E < \beta^2$.

## H.1. $T = 3$ case

We first consider the case where $T = 3$ and show that there is a continuous second-order phase transition as $\beta$ increases. We expand the energy function for $b$ near 0 to identify a critical point.

$$E^{\star}(\beta) = \beta^2 + \frac{1}{\beta^2} - \frac{(\beta b^2 + \frac{1}{\beta})^2}{1 + b^2 + b^4} \tag{103}$$

$$= \beta^2 + \frac{1}{\beta^2} - \frac{\frac{1}{\beta^2} + 2b^2 + k^2 b^4}{1 + b^2 + b^4} \tag{104}$$

Using a binomial approximation $(1 + b^2 + b^4)^{-1} \approx 1 - b^2 - b^4$,

$$E^{\star}(\beta) \approx \beta^2 + \frac{1}{\beta^2} - (\frac{1}{\beta^2} + 2b^2 + k^2 b^4)(1 - b^2 - b^4) \tag{105}$$

$$= \beta^2 + (\frac{1}{\beta^2} - 2)b^2 + (\frac{1}{\beta^2} + 2 - \beta^2)b^4 + (2 + \beta^2)b^6 + \beta^2 b^8 \tag{106}$$

which takes on the Landau form of $F(\beta, b) = F_0 + A(\beta)b^2 + \frac{B(\beta)}{2}b^4 + \ldots$. We identify a continuous second-order phase transition given by the quadratic coefficient $A(\beta) = \frac{1}{\beta^2} - 2$. Solving for the critical point where there is a sign change in the coefficient of $b^2$,

$$\beta = \frac{1}{\sqrt{2}} \tag{107}$$

We can see that at this critical point, $A(\beta) = 0$ and the system becomes unstable to fluctuations in $b$. Then, for $k > \frac{1}{\sqrt{2}}$, $A(\beta) < 0$ and the minimum of $E$ exists at some value $b \neq 0$. This also marks a transition in $u^{\star}$ due to its dependence on $b$.

## H.2. $T > 3$ case

Recall the effective energy function given by

$$E^{\star}(\beta) = \beta^2 + \frac{1}{\beta^2} - \frac{\frac{1}{\beta^2} + 2b^{T-1} + \beta^2 b^{2(T-1)}}{\sum_{i=0}^{T-1} b^{2i}} \tag{108}$$

We can rewrite the geometric sum $\sum_{i=0}^{T-1} b^{2i}$ as $\frac{1-b^{2T}}{1-b^2}$. Substituting in and expanding the terms yields

$$E^{\star}(\beta) = \beta^2 + \frac{1}{\beta^2} - \frac{(\frac{1}{\beta^2} - \frac{b^2}{\beta^2} + 2b^{T-1} - 2b^{T+1} + \beta^2 b^{2(T-1)} - \beta^2 b^{2T})}{1 - b^{2T}} \tag{109}$$

From this expression, we can observe that for $b$ near 0 (i.e., the denominator $1 - b^{2T} \approx 1$), the quadratic coefficient $A(\beta) = \frac{1}{\beta^2}$ is always positive, indicating that there is no small-$b$ instability and no second-order phase transition as $\beta$ changes. Hence, we instead consider the large $b$ case to study whether there is a point where large $b$ overtakes the trivial solution. Rewriting the energy function in terms of the dominating terms,

$$E^{\star}(\beta) \sim \beta^2 + \frac{1}{\beta^2} - \frac{-\beta^2 b^{2T}}{-b^{2T}} \tag{110}$$

$$= \frac{1}{\beta^2} \tag{111}$$

Thus for large $b$, the energy becomes $E = \frac{1}{\beta^2}$. Comparing to the energy for the trivial solution $E = \beta^2$, we see that for large $b$ and $\beta > 1$, the energy is lower than the trivial solution ($\frac{1}{\beta^2} < \beta^2$), indicating that the system has a first-order transition at $\beta = 1$ to nonzero $b$ and corresponding change in $u^{\star}$.

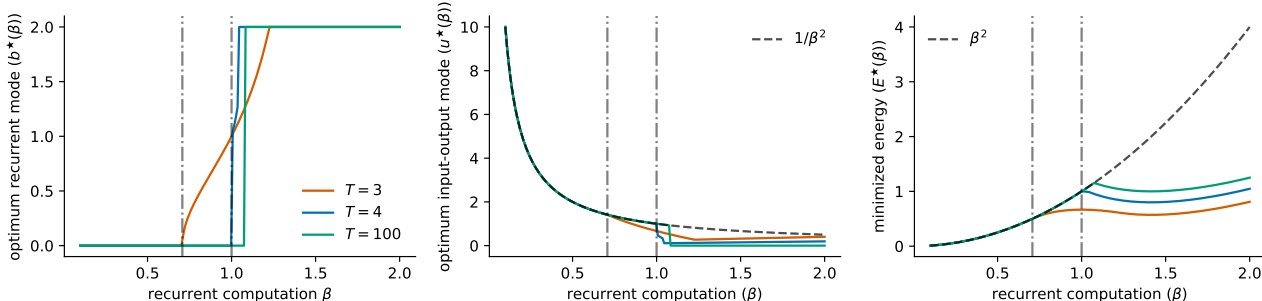

*Figure A3.* **Connectivity modes undergo a phase transition as the recurrent and feedforward computation is varied.** The change in (*left*) recurrent and (*center*) input-output connectivity modes as a function of $\beta$ are plotted, as well as (*right*) the minimum effective energy. The vertical dash-dot lines denote the critical points identified at $1/\sqrt{2}$ and 1. The black dashed line in the center figure denotes the trivial solution for the input-output modes at $u = \frac{1}{\beta^2}$ and in the right figure denotes the energy for the trivial solution at $E = \beta^2$.

We verify our work in simulation by plotting the minimizing recurrent modes and input-output modes, and the corresponding energy, for different values of $\beta$ in Figure A3. As predicted, a second-order phase transition in $b$ occurs at the critical point $\frac{1}{\sqrt{2}}$ for $T = 3$ and a first-order phase transition occurs near the critical point 1 for $T > 3$. This accompanied by the deviation of the input-output modes from the trivial solution $\frac{1}{\beta^2}$ and with an energy lower than the trivial solution $\beta^2$. For larger trajectory lengths, the simplifying approximations we make accumulate some error such that the critical point occurs at values slightly larger than 1, but we observe empirically that the critical point stabilizes around $\beta \approx 1.08$ across larger trajectory lengths.

This matches what we observe empirically in Figure 4, whereby the recurrent mode $b$ remains close to 0 and the input-output modes $ac$ match the singular value at the last timestep ($s_T = \kappa$) for small values of $s_1$ ($\beta$). As $\beta$ increases and the ratio of $s_1/s_T$ changes, the energy function reaches a critical point marking a first-order phase transition, where $b$ becomes nonzero, accompanied by a rapid change in $ac$.

Here we adopted the Dirac delta setting for mathematical simplicity, but we see that a similar phase transition of connectivity modes occurs for other task dynamics dependent on the relationship of the 'recurrent' computation (i.e., $s_{1:T-1}$) to the 'feedforward' computation ($s_T$) (Figure 4). This suggests an inherent interplay or trade-off between recurrent and feedforward computation in LRNNs (in the single-output case), dependent on the underlying task dynamics.

## I. Finite-width neural tangent kernel of LRNN

Here we derive the finite-width neural tangent kernel (NTK) (Jacot et al., 2018) for a LRNN where the loss is computed over the final output of the network, following a similar approach to Braun et al. (2022) in deep linear networks.

Recall the network function of the LRNN at training step $t_\theta$ is

$$\hat{\boldsymbol{Y}}_{T,t_\theta}(\boldsymbol{X}_{1:T}) = W_y \sum_{i=1}^{T} W_h^{T-i} W_x \boldsymbol{X}_i \tag{112}$$

After taking a training step with learning rate $\eta$, the network function becomes

$$\hat{\boldsymbol{Y}}_{T,t_\theta+1}(\boldsymbol{X}_{1:T}) = (W_y - \eta \frac{\partial \mathcal{L}}{\partial W_y}) \sum_{i=1}^{T} (W_h - \eta \frac{\partial \mathcal{L}}{\partial W_h})^{T-i} (W_x - \eta \frac{\partial \mathcal{L}}{\partial W_x}) \boldsymbol{X}_i \tag{113}$$

Using the binomial expansion $(a-b)^n = \sum_{k=0}^{n}(-1)^k \binom{n}{k}a^{n-k}b^k$,

$$(W_h - \eta\frac{\partial\mathcal{L}}{\partial W_h})^{T-i} = \sum_{k=0}^{T-i}(-1)^k\binom{T-i}{k}W_h^{T-i-k}(\eta\frac{\partial\mathcal{L}}{\partial W_h})^k \tag{114}$$

$$= W_h^{T-i} + \sum_{k=1}^{T-i}(-1)^k\binom{T-i}{k}W_h^{T-i-k}(\eta\frac{\partial\mathcal{L}}{\partial W_h})^k \tag{115}$$

Substituting back,

$$\hat{\boldsymbol{Y}}_{T,t_\theta+1}(\boldsymbol{X}_{1:T}) = (W_y - \eta\frac{\partial\mathcal{L}}{\partial W_y})\sum_{i=1}^{T}(W_h^{T-i} + \sum_{k=1}^{T-i}(-1)^k\binom{T-i}{k}W_h^{T-i-k}(\eta\frac{\partial\mathcal{L}}{\partial W_h})^k)(W_x - \eta\frac{\partial\mathcal{L}}{\partial W_x})\boldsymbol{X}_i \tag{116}$$

$$= \sum_{i=1}^{T}[W_y W_h^{T-i}W_x\boldsymbol{X}_i + W_y(\sum_{k=1}^{T-i}(-1)^k\binom{T-i}{k}W_h^{T-i-k}(\eta\frac{\partial\mathcal{L}}{\partial W_h})^k)W_x\boldsymbol{X}_i \tag{117}$$

$$- W_y\eta W_h^{T-i}\frac{\partial\mathcal{L}}{\partial W_x}\boldsymbol{X}_i - W_y\eta(\sum_{k=1}^{T-i}(-1)^k\binom{T-i}{k}W_h^{T-i-k}(\eta\frac{\partial\mathcal{L}}{\partial W_h})^k)\frac{\partial\mathcal{L}}{\partial W_x}\boldsymbol{X}_i \tag{118}$$

$$- \eta\frac{\partial\mathcal{L}}{\partial W_y}W_h^{T-i}W_x\boldsymbol{X}_i - \eta\frac{\partial\mathcal{L}}{\partial W_y}(\sum_{k=1}^{T-i}(-1)^k\binom{T-i}{k}W_h^{T-i-k}(\eta\frac{\partial\mathcal{L}}{\partial W_h})^k)W_x\boldsymbol{X}_i \tag{119}$$

$$+ \eta\frac{\partial\mathcal{L}}{\partial W_y}\eta W_h^{T-i}\frac{\partial\mathcal{L}}{\partial W_x}\boldsymbol{x}_i + \eta\frac{\partial\mathcal{L}}{\partial W_y}\eta(\sum_{k=1}^{T-i}(-1)^k\binom{T-i}{k}W_h^{T-i-k}(\eta\frac{\partial\mathcal{L}}{\partial W_h})^k)\frac{\partial\mathcal{L}}{\partial W_x}\boldsymbol{X}_i] \tag{120}$$

The gradient flow equation describing the dynamics of the network function is then

$$\frac{\hat{\boldsymbol{Y}}_{T,t_\theta+1} - \hat{\boldsymbol{Y}}_{T,t_\theta}}{\eta} = \sum_{i=1}^{T}[W_y(\sum_{k=1}^{T-i}(-1)^k\binom{T-i}{k}W_h^{T-i-k}(\eta)^{k-1}(\frac{\partial\mathcal{L}}{\partial W_h})^k)W_x\boldsymbol{X}_i - W_y W_h^{T-i}\frac{\partial\mathcal{L}}{\partial W_x}\boldsymbol{X}_i \tag{121}$$

$$- W_y(\sum_{k=1}^{T-i}(-1)^k\binom{T-i}{k}W_h^{T-i-k}(\eta\frac{\partial\mathcal{L}}{\partial W_h})^k)\frac{\partial\mathcal{L}}{\partial W_x}\boldsymbol{X}_i - \frac{\partial\mathcal{L}}{\partial W_y}\sum_{i=1}^{T}W_h^{T-i}W_x\boldsymbol{X}_i \tag{122}$$

$$- \frac{\partial\mathcal{L}}{\partial W_y}(\sum_{k=1}^{T-i}(-1)^k\binom{T-i}{k}W_h^{T-i-k}(\eta\frac{\partial\mathcal{L}}{\partial W_h})^k)W_x\boldsymbol{x}_i + \frac{\partial\mathcal{L}}{\partial W_y}\eta W_h^{T-i}\frac{\partial\mathcal{L}}{\partial W_x}\boldsymbol{X}_i \tag{123}$$

$$+ \frac{\partial\mathcal{L}}{\partial W_y}\eta(\sum_{k=1}^{T-i}(-1)^k\binom{T-i}{k}W_h^{T-i-k}(\eta\frac{\partial\mathcal{L}}{\partial W_h})^k)\frac{\partial\mathcal{L}}{\partial W_x}\boldsymbol{X}_i] \tag{124}$$

As the learning rate $\eta \to 0$ (the gradient flow regime),

$$\tau\frac{\hat{\boldsymbol{Y}}_T}{dt_\theta} = \sum_{i=1}^{T}(-W_y W_h^{T-i}\frac{\partial\mathcal{L}}{\partial W_x} - (T-i)W_y W_h^{T-i-1}\frac{\partial\mathcal{L}}{\partial W_h}W_x - \frac{\partial\mathcal{L}}{\partial W_y}W_h^{T-i}W_x)\boldsymbol{X}_i \tag{125}$$

Substituting the partial derivatives of the loss,

$$\tau\frac{\hat{\boldsymbol{Y}}_T}{dt_\theta} = \sum_{i=1}^{T}-W_y W_h^{T-i}(\sum_{j=1}^{T}W_h^{(T-j)\top}W_y^\top(\boldsymbol{Y}_T - W_y\sum_{k=1}^{T}W_h^{T-k}W_x\boldsymbol{X}_k)\boldsymbol{X}_j^\top)\boldsymbol{X}_i \tag{126}$$

$$- (T-i)W_y W_h^{T-i-1}(\sum_{j=1}^{T-1}\sum_{r=0}^{T-j-1}W_h^{(r)\top}W_y^\top(\boldsymbol{Y}_T - W_y\sum_{k=1}^{T}W_h^{T-k}W_x\boldsymbol{X}_k)\boldsymbol{X}_j^\top W_x^\top W_h^{(T-j-1-r)\top}W_x\boldsymbol{X}_i \tag{127}$$

$$- ((\boldsymbol{Y}_T - W_y\sum_{k=1}^{T}W_h^{T-k}W_x\boldsymbol{X}_k)(\sum_{j=1}^{T}\boldsymbol{X}_j^\top W_x^\top W_h^{(T-j)\top}))W_h^{T-i}W_x\boldsymbol{X}_i \tag{128}$$

Finally, we use the identity $\text{vec}(AXB) = (B^\top \otimes A)\text{vec}(X)$ to derive the NTK $(\nabla_\theta \text{vec}(\hat{Y}_T) \nabla_\theta \text{vec}(\hat{Y}_T))$ on the left-side of the vectorizing function

$$\tau \frac{d\text{vec}(\hat{Y}_T)}{dt_\theta} = (\sum_{i=1}^{T} \sum_{j=1}^{T-1} -X_i^\top X_j \otimes W_y W_h^{T-i} W_h^{(T-j)\top} W_y^\top \tag{129}$$

$$-[\sum_{r=0}^{T-i-1} -(T-i) X_i^\top W_x^\top W_h^{(T-k-1-r)} W_x X_j \otimes W_y W_h^{T-i-1} W_h^{(r)\top} W_y^\top] \tag{130}$$

$$- I_{N_y} \otimes X_i^\top W_x^\top W_h^{(T-i)\top} W_h^{(T-j)} W_x X_j) \text{vec}(Y_T - W_y \sum_{k=1}^{T} W_h^{T-k} W_x X_k) \tag{131}$$

$$\nabla_\theta \text{vec}(\hat{Y}_T) \nabla_\theta \text{vec}(\hat{Y}_T) = (\sum_{i=1}^{T} \sum_{j=1}^{T-1} X_i^\top X_j \otimes W_y W_h^{T-i} W_h^{(T-j)\top} W_y^\top \tag{132}$$

$$+[\sum_{r=0}^{T-i-1} -(T-i) X_i^\top W_x^\top W_h^{(T-k-1-r)} W_x X_j \otimes W_y W_h^{T-i-1} W_h^{(r)\top} W_y^\top] \tag{133}$$

$$+ I_{N_y} \otimes X_i^\top W_x^\top W_h^{(T-i)\top} W_h^{(T-j)} W_x X_j) \tag{134}$$

## J. Analyzing the impact of recurrence on feature learning

Here we study how increasing trajectory length $T$ (becoming more 'recurrent') and the scale of initialization weights influence the learning dynamics of the network, as measured by the movement of the NTK and illustrated in Figure 5. We conduct several different experiments in networks with aligned and unaligned weights and observe that both settings yield similar qualitative behavior. Networks with unaligned weights are initialized with weights drawn from a normal distribution centered at 0 with a specified variance (initialization scale), and networks with aligned weights have singular vectors (or eigenvectors for the rotational task below) aligned with the data correlation matrix singular vectors (eigenvectors) and connectivity modes initialized at the corresponding initialization scale. Hence, both settings have the same expected spectral radius (equal to the initialization scale), but different means and variance (unaligned: mean is 0, variance is initialization scale; aligned: mean is initialization scale, variance is 0). During training, we calculate the kernel distance of the NTK at training step $t_\theta$ ($K(t_\theta)$) from its initialization ($K(0)$) to quantify feature learning. Following the definition in Fort et al. (2020), the kernel distance $D(t_\theta)$ is defined as

$$D(t_\theta) = 1 - \frac{\langle K(0), K(t_\theta) \rangle}{\|K(0)\|_F \|K(t_\theta)\|_F}, \tag{135}$$

We visualize the kernel distance between initialization and the end of training as heatmaps, varying the trajectory length and the initialization scale to study how these two factors impact the NTK. In the main text (Figure 5), we show the case where networks are trained to perform perfect integration (summation) of input (i.e., constant task dynamics where $\lambda = 1, \delta = 1, s_t = 1$). For completeness, in Figure A4 we also show networks trained on rotational task dynamics of angle $\phi_\lambda = \pi/12$ and radius $R_\lambda = 1$ (i.e., inverse-exponential task dynamics where $\lambda = 1e^{\frac{\pi}{12}i}, \delta = 1, d_t = \delta\lambda^{T-t}$; see Appendices N and O for details on rotational task dynamics).

We consistently see that recurrence ($T > 1$) leads to kernel movement across a large range of initialization strengths which contrasts with feedforward networks ($T = 1$). This occurs in both aligned and unaligned networks.

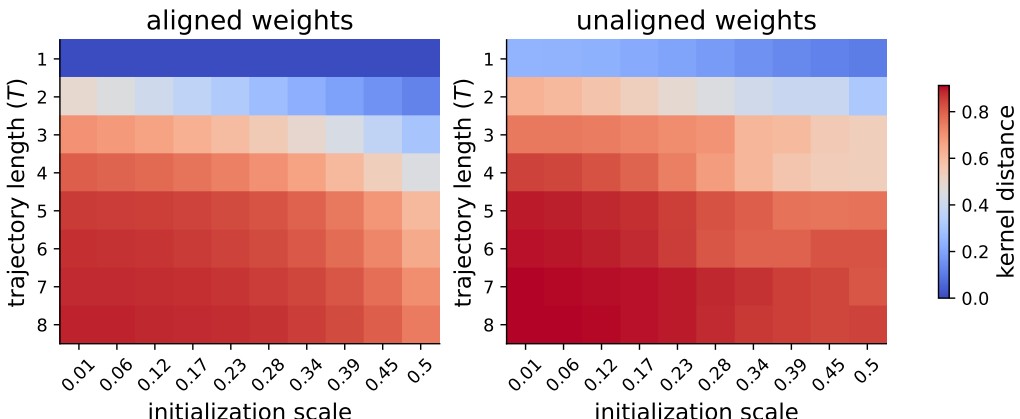

*Figure A4.* **NTK distance is qualitatively similar in rotational task.** Phase plots illustrating the kernel distance of the NTK from initialization to the end of training. Networks are trained to perform rotations of angle $\phi_\lambda = \pi/12$ and show qualitatively similar behavior as networks trained on constant task dynamics in Figure 5.

**Infinite width.**  As stated in the main text, a network's learning regime is determined by its initialization scale and its hidden layer width. It's known that as networks increase in width (infinite-width limit) and/or initialization scale of the weights, feature learning is reduced and instead lazy learning occurs. Since our first experiments (Figures 5 and A4) show that the NTK changes more as networks transition from feedforward to recurrent networks, a natural next question is whether lazy learning is still induced by large initializations and network widths in recurrent networks. To study the impact of increasing network width, we repeat the same experiment on networks with unaligned weights trained on constant task dynamics (Figure 5), keeping all parameters identical except for the hidden layer size, which is set to 300 (rather than 4 as before), while the task itself only spans 4 singular value dimensions. While the RNNs we study are still small compared to the size of networks used for typical applications, they are still significantly overparameterized relative to the task we train on and serve as a qualitative comparison between different network widths. In Figure A5, we see that, although there is still substantial kernel movement, especially as trajectory length increases, this occurs primarily at smaller initializations compared to the prior experiments. Although we only study a network of size 300, it's possible that this pattern continues with larger network widths (Alemohammad et al., 2021), as in feedforward networks.

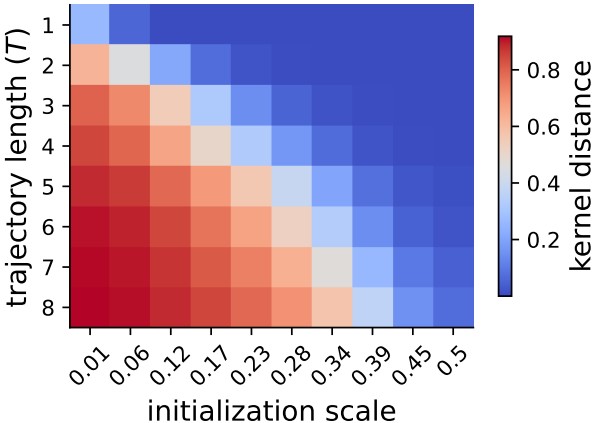

*Figure A5.* **Effects of recurrence on NTK movement are reduced with increasing width.** Phase plots illustrating the kernel distance of the NTK from initialization when the hidden layer size is set to 300 compared to size 4 in LRNNs with unaligned weights. There is still movement in the NTK, but at smaller initialization scales compared to networks with smaller widths.

**Large initialization scale.** Next, we study the effect of larger initializations (which typically induce lazy learning) on kernel distance. This is more challenging to do in recurrent networks than feedforward networks because of the instability of training as initializations approach 1. However, networks with aligned weights are comparatively more stable during training because their optimization is slightly simplified from its initialization. We show the results in Figure A6. As we can see, the effect of longer trajectory length yielding greater kernel distance persists across large initializations as well, especially in unaligned networks. However, we also see, especially in aligned networks, that this kernel distance decreases as initialization approaches 1. This is expected. Under the aligned configuration, (singular/eigen-) vector rotation (which typically occurs early in training in unaligned networks in the rich learning regime) to align with the task is unnecessary and only the scaling from initialization changes. In this case, when the network's initialization is close to the target value, the adjustments required to fit the target are minimized and the kernel distance is smaller. In the tasks we study here, the target has a magnitude of 1, explaining why smaller initialization scales result in more pronounced NTK movement and why initializations close to 1 result in minimal movement for aligned networks. In contrast, unaligned networks still have pronounced kernel distance at large initializations likely because vector rotation still substantially changes the NTK beyond scaling of (singular/eigen-) values. The fact that kernel distance still substantially changes at large initialization scales is quite surprising, but supports the idea that the optimization of the recurrent layer might force some form of feature learning. In parallel with the publication of this work, Bordelon et al. (2025) has shown how weight initialization scale and outlier eigenvalues affect feature learning.

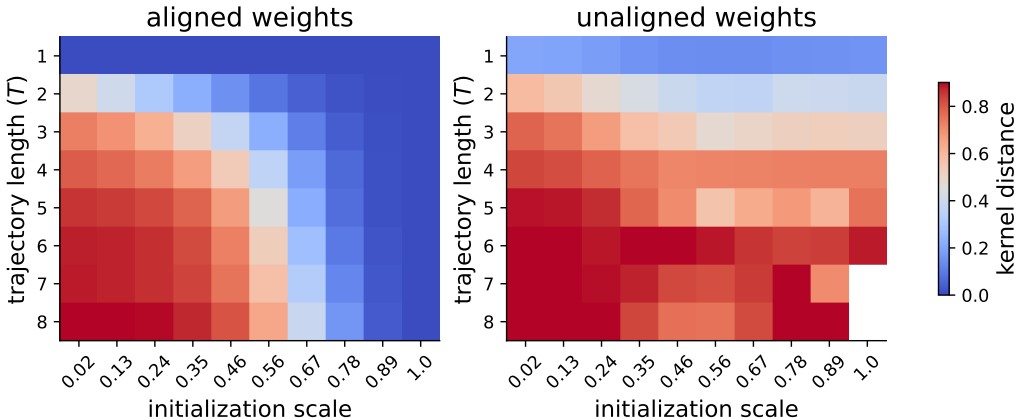

*Figure A6.* **Effects of recurrence on NTK movement persist across larger initializations.** Phase plots illustrating the kernel distance of the NTK from initialization across larger initializations for (*left*) aligned and (*right*) unaligned weights. The white boxes indicate NaNs. As expected, networks with aligned weights decrease in kernel distance as their initialization approaches the target value at 1 because there is no (singular/eigen-) vector alignment occurring and the change in parameters to fit the target is minimal. However, networks with unaligned weights still exhibit substantial kernel movement even for large weight initializations, which is particularly surprising.

.

**Independently initialized modes** Our final experiment studies how the initialization of input-output modes and recurrent modes affect kernel distance, and whether there is any meaningful difference between them. This experiment is motivated by Schuessler et al. (2024), which found that the initialization strength of the readout weight (analogous to output modes in our case) acts as a control parameter in RNNs that induces either aligned ('rich') dynamics, or oblique ('lazy') dynamics. Although we do not explicitly study the alignment of the dynamics (note that aligned dynamics differs from what we here refer to as aligned weights), we are interested in seeing how the scaling of the input-output mode might impact kernel distance when varied independently from the recurrent mode. We note, however, that our setting differs from Schuessler et al. (2024) in that we study linear RNNs, which might substantially change network behavior.

Interestingly, by independently varying the initialization strength of the input-output modes and recurrent modes, we see that the initialization of the recurrent mode more strongly impacts kernel distance than the input-output modes– the recurrent mode mostly determines the NTK distance. Intuitively, this seems reasonable given that the NTK distance increases with trajectory length (corresponding to the 'recurrent mode' entering into the function performed by the network more). This result also supports the idea that the increase in NTK distance with trajectory length is driven by the repeated application of the recurrent mode and modulated by its initialization. We suggest that the constrained optimization in RNNs might

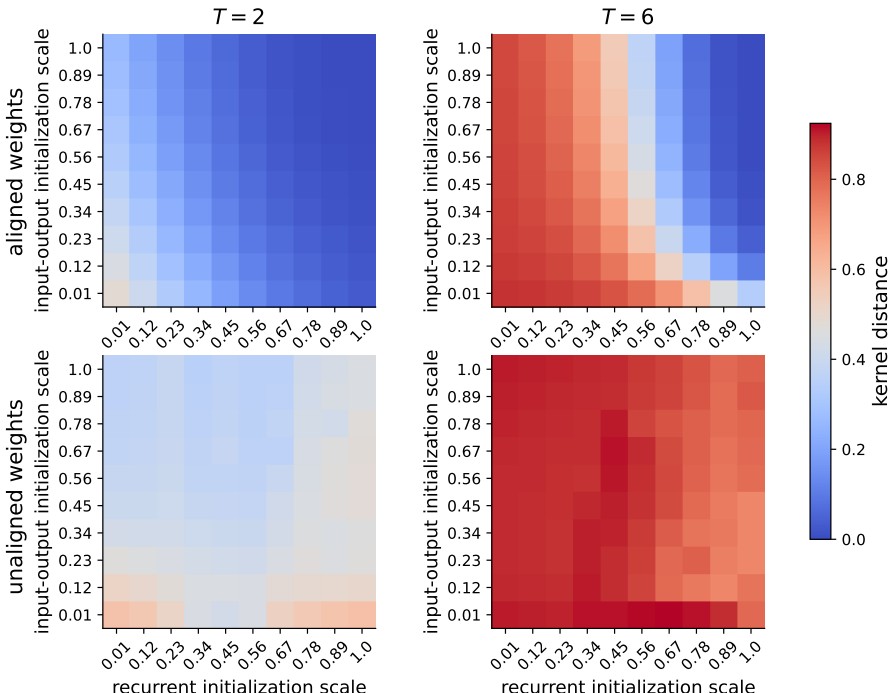

*Figure A7.* **Initialization of recurrent modes more strongly impacts NTK movement.** Phase plots illustrating the kernel distance of the NTK from initialization for different trajectory lengths ((*left*) $T = 2$, (*right*) $T = 6$) as a function of initialization strength of input-output and recurrent connectivity modes, which are independently varied. Visualized for RNNs with (*top*) aligned and (*bottom*) unaligned weights.

encourage or force feature learning in recurrent modes, but more work is needed to understand this. In Appendix K, we show that the energy function is impacted differently by errors in recurrent connectivity modes versus input-output modes, which might provide some intuition for the differences we discuss here.

A note of caution that the continuous increase in kernel distance with trajectory length in these experiments does not necessarily indicate that the network continues becoming a 'richer' learner as $T \to \infty$, but is likely, at least in part, due to the fact that changes in the weights are amplified more by the NTK for longer trajectory lengths. This makes it more challenging to compare NTK movement across different trajectory lengths. Nevertheless, the NTK is still non-constant in the cases with non-zero kernel distance, indicating some feature learning is occurring.

Returning back to Schuessler et al. (2024), we see that in our setting the initialization of the input-output mode does not appear to substantially affect the NTK behavior. One possible reason may simply be that this effect arises specifically in nonlinear networks, or in networks trained on particular tasks (not constant integration as we consider here). It might also be the case that the distinction between aligned and oblique dynamics are not adequately captured by the NTK distance, and that separate analyses are required to evaluate these differences. We leave this to future work.

## K. Impact of connectivity modes on the energy function

Here, we analyze how the energy function is impacted differently by errors in the recurrent connectivity mode versus the input-output connectivity modes.

Recall the energy function is given by

$$E = \frac{1}{2\tau} \sum_{i=1}^{T} (s_i - cb^{T-i}a)^2 \tag{136}$$

and that we can decompose the data singular values as $s_t = \delta f(\lambda, t)$. We make a change of variables such that $ac = \delta + \Delta_{ac}$ and $b = f(\lambda, i)^{1/(T-i)} + \Delta_{b_i}$. When $\Delta_{ac} = \Delta_{b_i} = 0 \ \forall i$, $ac, b$ are at the global solution. Substituting into the energy

function,

$$E = \frac{1}{2\tau} \sum_{i=1}^{T} (\delta f(\lambda, i) - (\delta + \Delta_u)(f(\lambda, i)^{1/(T-i)} + \Delta_{b_i})^{T-i})^2 \tag{137}$$

To analyze the effect of error in the input-output modes, we first consider the case where $\Delta_{b_i} = 0 \ \forall i$.

$$E = \frac{1}{2\tau} \sum_{i=1}^{T} (\delta f(\lambda, i) - (\delta + \Delta_{ac}) f(\lambda, i))^2 \tag{138}$$

$$= \frac{1}{2\tau} \sum_{i=1}^{T} (-\Delta_{ac} f(\lambda, i))^2 \tag{139}$$

We can see that the error of the input-output connectivity modes $\Delta_{ac}$ enters into the energy function quadratically. Instead, for the case where $\Delta_{ac} = 0$ and $\Delta_{b_i} \neq 0$,

$$E = \frac{1}{2\tau} \sum_{i=1}^{T} (\delta f(\lambda, i) - \delta (f(\lambda, i)^{1/(T-i)} + \Delta_{b_i})^{T-i})^2 \tag{140}$$

$$= \frac{1}{2\tau} \sum_{i=1}^{T} (\delta f(\lambda, i) - \delta (\sum_{k=0}^{T-i} \binom{T-i}{k} (f(\lambda, i)^{1/(T-i)})^{T-i-k} \Delta_{b_i}^{k}))^2 \tag{141}$$

$$= \frac{1}{2\tau} \sum_{i=1}^{T} (\delta f(\lambda, i) - \delta (f(\lambda, i) + \sum_{k=1}^{T-i} \binom{T-i}{k} (f(\lambda, i)^{1/(T-i)})^{T-i-k} \Delta_{b_i}^{k}))^2 \tag{142}$$

$$= \frac{1}{2\tau} \sum_{i=1}^{T} (-\delta \sum_{k=1}^{T-i} \binom{T-i}{k} (f(\lambda, i)^{1/(T-i)})^{T-i-k} \Delta_{b_i}^{k})^2 \tag{143}$$

Here we can see that the error of the recurrent connectivity mode $\Delta_{b_i}$ enters into the energy function exponentially with time.

## L. Early-importance task dynamics lead to unstable solutions

We extend our sensory integration task in the main text to consider the impact of increasing versus decreasing task dynamics on stability of the network. Similar to before, we remove prior assumptions about whitened data and aligned weights, and show that our predictions made in aligned linear RNNs generalize. Our setting is identical to that in Section 3.6 except that rather than taking the output to be the mean or the sum of the input in each dimension, we instead vary the relation to be scaled with increasing weight over time (late-importance: $\boldsymbol{y}_T = 0.1 \sum_{t=1}^{T} t\boldsymbol{x}_t$) or decreasing weight over time (early-importance: $\boldsymbol{y}_T = \sum_{t=1}^{T} \frac{1}{t}\boldsymbol{x}_t$). We select tasks that do not have inverse-exponential dynamics to illustrate the generality of our theory (and also verify that this behavior occurs with inverse-exponential dynamics). Our theory predicts that early-importance task dynamics (if the scaling term reflected in the input-output modes is sufficiently large) are *unstable*. To visualize stability of learning, we plot the gradient of the hidden layer ($\frac{d\boldsymbol{H}_T}{dt_\theta}$) throughout training. We see that in the late-importance case, the network singular values converge to the same solutions in each dimension and the hidden layer gradient remains small ($< 10^{-3}$). Conversely, for early-importance task dynamics, we observe that optimization is more challenging as only some of the network singular values converge to the solution and others fluctuate at local minima. In parallel, the hidden layer gradient is larger ($\sim 0.3$) and fluctuates throughout training. We are only able to visualize this effect because we study a network trained on a trajectory length of 6 ($T = 6$): for large $T$, exploding gradients make training impossible.

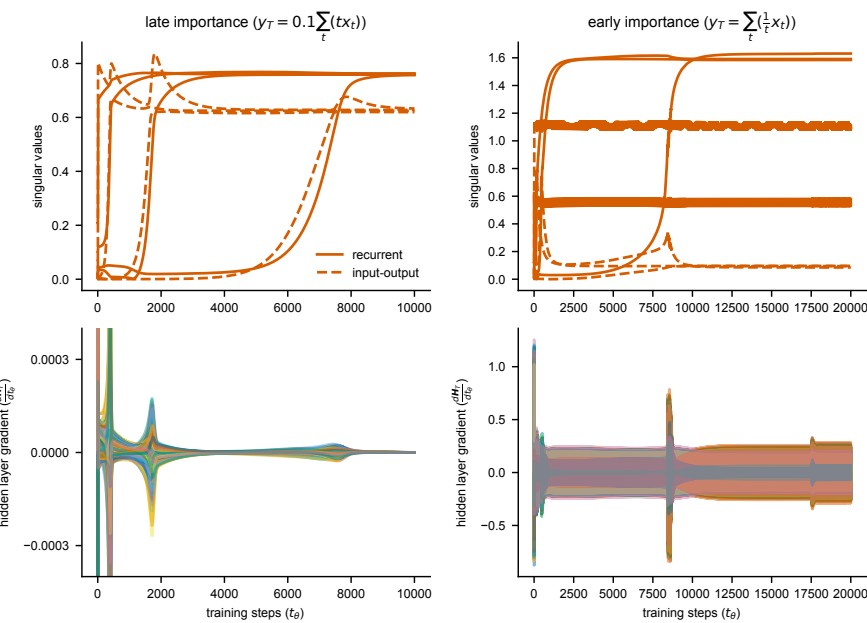

*Figure A8.* **Early-importance task dynamics yield unstable solutions.** We train LRNNs with small, random weights on either a task with (*left*) late-importance dynamics ($\boldsymbol{y}_T = 0.1\sum_t(t\boldsymbol{x}_t)$) or (*right*) early-importance dynamics ($\boldsymbol{y}_T = \sum_t(\frac{1}{t}\boldsymbol{x}_t)$). *(Top)* All of the LRNN singular values converge to the same solution when task dynamics are stable (late-importance), but not when they are unstable (early-importance). *(Bottom)* The hidden layer gradient illustrates the (in)stability of each network, smoothly converging to 0 through training with small magnitude in the late-importance case and fluctuating throughout training with large magnitude in the early-importance case. .

## M. Extending to the (autoregressive) $T$-output case

Although our main analyses in this work focus on the single-output case for simplicity, here we extend our main theoretical framework to the $T$-output case for completeness. The network receives an input $\boldsymbol{x}_t \in \mathbb{R}^{N_x}$ at each timestep $t$ and produces an output at every timestep $\hat{\boldsymbol{y}}_t \in \mathbb{R}^{N_y}$. In this case, the loss is computed over the output at every timestep over $P$ trajectories $\{\boldsymbol{x}_{p,1}, \boldsymbol{x}_{p,2}, \ldots, \boldsymbol{x}_{p,T}, \boldsymbol{y}_{p,1}, \boldsymbol{y}_{p,2}, \ldots, \boldsymbol{y}_{p,T}\}$

$$\mathcal{L} = \frac{1}{2}\sum_{p=1}^{P}\sum_{t=1}^{T}\|\boldsymbol{y}_{p,t} - W_y(\sum_{i=1}^{t}W_h^{t-i}W_x\boldsymbol{x}_{p,i})\|^2 \tag{144}$$

The derivation is the same as the single-output case, except that now, there is an addition summation over the input-output correlation matrices for the output at different timesteps

$$\Sigma^{Y_t X_{t'}} = \sum_{p=1}^{P}\boldsymbol{y}_{p,t}\boldsymbol{x}_{p,t'}^{\top} \tag{145}$$

Under the same assumptions as before of whitened input with 0 mean and the gradient flow regime, the gradient equations

become

$$\tau \frac{d}{dt_\theta} W_x = \sum_{t=1}^{T} \sum_{i=1}^{t} W_h^{(t-i)\top} W_y^\top (\Sigma^{Y_t X_i} - W_y W_h^{t-i} W_x) \tag{146}$$

$$\tau \frac{d}{dt_\theta} W_h = \sum_{t=1}^{T} \sum_{i=1}^{t-1} \sum_{r=0}^{t-i-1} W_h^{(r)\top} W_y^\top (\Sigma^{Y_t X_i} - W_y W_h^{t-i} W_x) W_x^\top W_h^{(t-i-1-r)\top} \tag{147}$$

$$\tau \frac{d}{dt_\theta} W_y = \sum_{t=1}^{T} \sum_{i=1}^{t} (\Sigma^{Y_t X_i} - W_y W_h^{t-i} W_x) W_x^\top W_h^{(t-i)\top} \tag{148}$$

Again, assuming the input-output correlation matrices have constant left and right singular vectors across trajectory timesteps and that the LRNN is data-aligned at initialization, we get diagonal matrices $\overline{W}_x, \overline{W}_h, \overline{W}_y$,

$$\tau \frac{d}{dt_\theta} \overline{W}_x = \sum_{t=1}^{T} \sum_{i=1}^{t} \overline{W}_h^{(t-i)\top} \overline{W}_y^\top (S_{t,i} - \overline{W}_y \overline{W}_h^{t-i} \overline{W}_x) \tag{149}$$

$$\tau \frac{d}{dt_\theta} \overline{W}_h = \sum_{t=1}^{T} \sum_{i=1}^{t-1} \sum_{r=0}^{t-i-1} \overline{W}_h^{(r)\top} \overline{W}_y^\top (S_{t,i} - \overline{W}_y \overline{W}_h^{t-i} \overline{W}_x) \overline{W}_x^\top \overline{W}_h^{(t-i-1-r)\top} \tag{150}$$

$$\tau \frac{d}{dt_\theta} \overline{W}_y = \sum_{t=1}^{T} \sum_{i=1}^{t} (S_{t,i} - \overline{W}_y \overline{W}_h^{t-i} \overline{W}_x) \overline{W}_x^\top \overline{W}_h^{(t-i)\top} \tag{151}$$

where $S_{t,i}$ is the singular value matrix of the input-output correlation matrix between $Y_t$ at trajectory time $t$ and $X_i$ at trajectory time $i$.

Rewriting the equations in terms of the decoupled connectivity modes along each singular value dimension

$$\tau \frac{d}{dt_\theta} a_\alpha = \sum_{t=1}^{T} \sum_{i=1}^{t} b_\alpha^{t-i} c_\alpha (s_{\alpha,t,i} - c_\alpha b_\alpha^{t-i} a_\alpha) \tag{152}$$

$$\tau \frac{d}{dt_\theta} b_\alpha = \sum_{t=1}^{T} \sum_{i=1}^{t-1} \sum_{r=0}^{t-i-1} b_\alpha^{(r)} c_\alpha (s_{\alpha,t,i} - c_\alpha b_\alpha^{t-i} a_\alpha) a_\alpha b_\alpha^{(t-i-1-r)} \tag{153}$$

$$= \sum_{t=1}^{T} \sum_{i=1}^{t-1} (t-i) c_\alpha (s_{\alpha,t,i} - c_\alpha b_\alpha^{t-i} a_\alpha) a_\alpha b_\alpha^{(t-i-1)} \tag{154}$$

$$\tau \frac{d}{dt_\theta} c_\alpha = \sum_{t=1}^{T} \sum_{i=1}^{t} (s_{\alpha,t,i} - c_\alpha b_\alpha^{t-i} a_\alpha) a_\alpha b_\alpha^{t-i} \tag{155}$$

These dynamics arise from gradient descent on the energy function

$$E = \frac{1}{2\tau} \sum_\alpha \sum_{t=1}^{T} \sum_{i=1}^{t} (s_{\alpha,t,i} - c_\alpha b_\alpha^{t-i} a_\alpha)^2 \tag{156}$$

Here we discuss the extension of some of our results to the autoregressive case, although we note that fully characterizing the network behavior in this setting will be an important direction for future work.

## M.1. Exact solution of input-output connectivity modes

We can apply the same approach as in Appendix C to get the exact solutions of the input-output connectivity modes in the autoregressive case, which is given by:

$$u(t_\theta) = \frac{e^{2t_\theta(\sum_{t=1}^{T} \sum_{i=1}^{t} b^{t-i} s_{t,i})/\tau}(\sum_{t=1}^{T} \sum_{i=1}^{t} b^{t-i} s_{t,i})}{(\sum_{t=1}^{T} \sum_{i=1}^{t} b^{t-i} s_{t,i})/u(0) - (\sum_{t=1}^{T} \sum_{i=1}^{t} b^{2(t-i)}) + e^{2t_\theta(\sum_{t=1}^{T} \sum_{i=1}^{t} b^{t-i} s_{t,i})/\tau}(\sum_{t=1}^{T} \sum_{i=1}^{t} b^{2(t-i)})} \tag{157}$$

## M.2. Local approximation of recurrent connectivity modes

Again, we can apply the same approach as in Appendix D, performing a Taylor expansion of the learning dynamics of the recurrent mode. The equations in Appendix D remain identical except for the $n$th partial derivative of the energy function with respect to the recurrent mode $b$, which is given by the following in the autoregressive case:

$$\frac{d^n E}{db^n} = \sum_{t=1}^{T} \sum_{i=1}^{t-n} \left[ (t-i) \left( \prod_{j=1}^{n-1} (t-i-j) \right) s_{t,i} cab^{t-i-n} \right] - \sum_{t=1}^{T} \sum_{i=1}^{(2t-n)/2} \left[ (t-i) \left( \prod_{j=1}^{n-1} (2t-2i-j) \right) c^2 a^2 b^{2t-2i-n} \right]$$

(158)

By substituting this form of $\frac{d^n E}{db^n}$, the approximation of the recurrent learning dynamics can be computed for the autoregressive case.

## M.3. Zero-loss solutions only exist for inverse-exponential task dynamics

The autoregressive case has an energy function that is very similar to the single-output case. In this case, following the same logic as Appendix F, zero-loss solutions only exist for inverse-exponential task dynamics given by the form $s_{t,i} = \delta \lambda^{t-i} \forall t, i$. This has its global solution at $b = \lambda, ac = \delta$.

## M.4. Existence of effective regularization term

By decomposing the energy function in the autoregressive case here, we can see that an effective regularization term appears as before, albeit in a different form:

$$E = \frac{1}{2\tau} \underbrace{\left( \sum_{t=1}^{T} \sum_{i=1}^{t} s_{t,i}^2 - 2 s_{t,i} c b^{t-i} a \right)}_{\text{data-driven term}} + \underbrace{\frac{1}{2\tau} c^2 a^2 \frac{T(1-b^2) - b^2(1-b^{2T})}{(1-b^2)^2}}_{\text{effective regularization term}}$$

(159)

## M.5. Neural tangent kernel

Using the same approach as in Appendix I, the NTK is given by:

$$\nabla_\theta \text{vec}(\hat{Y}_t) \nabla_\theta \text{vec}(\hat{Y}_t) = \left( \sum_{t=1}^{T} \sum_{i=1}^{t} \sum_{j=1}^{t-1} \boldsymbol{X}_i^\top \boldsymbol{X}_j \otimes W_y W_h^{t-i} W_h^{(t-j)\top} W_y^\top \right. \tag{160}$$

$$+ \left[ \sum_{r=0}^{t-i-1} -(t-i) \boldsymbol{X}_i^\top W_x^\top W_h^{(t-k-1-r)} W_x \boldsymbol{X}_j \otimes W_y W_h^{t-i-1} W_h^{(r)\top} W_y^\top \right] \tag{161}$$

$$\left. + I_{N_y} \otimes \boldsymbol{X}_i^\top W_x^\top W_h^{(t-i)\top} W_h^{(t-j)} W_x \boldsymbol{X}_j \right) \tag{162}$$

# N. Generalizing gradient flow equations to the eigenspace to capture rotations

In the main body of the paper, we study learning dynamics in the parameter singular value space, assuming that the network aligns itself with the input-output correlation matrices' singular vectors. This has the advantage of allowing for different input and output sizes and is also easier to analyze because singular values are real and nonnegative. However, one notable limitation of studying the learning dynamics in the singular value space is that it restricts expressivity of the network. In particular, the assumption of constant left and right singular vectors, necessary to simplify the gradient forms and arrive at decoupled connectivity modes, restricts the RNN to unidirectional scaling and summation (i.e., integration) of input.

In this section, we derive our main gradient equations in the eigenspace to capture the other primary computation a linear RNN can perform: rotations (oscillations). Notably, this derivation allows for complex eigenvalues and still has decoupled gradient dynamics across different dimensions. We make the same assumptions on the input as before, namely that inputs are whitened with 0 mean, and we assume the gradient flow regime. The derivation remains the same as in Appendix B until

Equation (23), which we copy here for convenience:

$$\tau \frac{d}{dt_\theta} W_x = \sum_{i=1}^{T} W_h^{(T-i)\top} W_y^\top (\Sigma^{YX_i} - W_y W_h^{T-i} W_x) \tag{163}$$

$$\tau \frac{d}{dt_\theta} W_h = \sum_{i=1}^{T-1} \sum_{r=0}^{T-i-1} W_h^{(r)\top} W_y^\top (\Sigma^{YX_i} - W_y W_h^{T-i} W_x) W_x^\top W_h^{(T-i-1-r)\top} \tag{164}$$

$$\tau \frac{d}{dt_\theta} W_y = \sum_{i=1}^{T} (\Sigma^{YX_i} - W_y W_h^{T-i} W_x) W_x^\top W_h^{(T-i)\top} \tag{165}$$

Whereas before in Appendix B we assumed that the input-output correlation matrices $\Sigma^{YX_t}$ have constant left and right singular vectors, here we instead assume that the input-output correlation matrices are normal and have constant eigenvectors across trajectory timesteps. This is a more general assumption than that of constant left and right singular vectors, because it allows for complex eigenvalues and thus captures rotational teachers.

*Proof.* Data generated by a linear RNN teacher parameterized by diagonalizable matrices $\tilde{W}_x, \tilde{W}_h, \tilde{W}_y$ with the same eigenvectors $P$ have constant eigenvectors across trajectory timesteps.

$$\Sigma^{YX_t} = \sum_{p=1}^{P} \boldsymbol{y}_{p,T} \boldsymbol{x}_{p,t}^\top \tag{166}$$

$$= \sum_{p=1}^{P} \tilde{W}_y \tilde{W}_h^{T-t} \tilde{W}_x \boldsymbol{x}_{p,t} \boldsymbol{x}_{p,t}^\top \tag{167}$$

$$= \tilde{W}_y \tilde{W}_h^{T-t} \tilde{W}_x \tag{168}$$

$$= P D_y P^\dagger (P D_h P^\dagger)^{T-t} P D_x P^\dagger \tag{169}$$

$$= P D_y D_h D_x P^\dagger \tag{170}$$

$$= P D_t P^\dagger \tag{171}$$

where $P^\dagger$ denotes the conjugate transpose of $P$. $\qquad \square$

Substituting the diagonalized form of the data-correlation matrix $\Sigma^{YX_t}$ into the gradient flow equations yields

$$\tau \frac{d}{dt_\theta} W_x = \sum_{i=1}^{T} W_h^{(T-i)\top} W_y^\top (P D_i P^\dagger - W_y W_h^{T-i} W_x) \tag{172}$$

$$\tau \frac{d}{dt_\theta} W_h = \sum_{i=1}^{T-1} \sum_{r=0}^{T-i-1} W_h^{(r)\top} W_y^\top (P D_i P^\dagger - W_y W_h^{T-i} W_x) W_x^\top W_h^{(T-i-1-r)\top} \tag{173}$$

$$\tau \frac{d}{dt_\theta} W_y = \sum_{i=1}^{T} (P D_i P^\dagger - W_y W_h^{T-i} W_x) W_x^\top W_h^{(T-i)\top} \tag{174}$$

For a real matrix $A$, $A^\top = A^\dagger$. Using this, we can rewrite the transposed parameter matrices above as

$$\tau \frac{d}{dt_\theta} W_x = \sum_{i=1}^{T} W_h^{(T-i)\dagger} W_y^\dagger (P D_i P^\dagger - W_y W_h^{T-i} W_x) \tag{175}$$

$$\tau \frac{d}{dt_\theta} W_h = \sum_{i=1}^{T-1} \sum_{r=0}^{T-i-1} W_h^{(r)\dagger} W_y^\dagger (P D_i P^\dagger - W_y W_h^{T-i} W_x) W_x^\dagger W_h^{(T-i-1-r)\dagger} \tag{176}$$

$$\tau \frac{d}{dt_\theta} W_y = \sum_{i=1}^{T} (P D_i P^\dagger - W_y W_h^{T-i} W_x) W_x^\dagger W_h^{(T-i)\dagger} \tag{177}$$

$$\tag{178}$$

As before, we assume the LRNN is *data-aligned* at initialization, but in the *eigenspace*, such that the network has the same eigenvectors as the data correlation eigenvectors $P$. More specifically, $W_x(0) = P\overline{W}_x(0)P^\dagger$, $W_h(0) = P\overline{W}_h(0)P^\dagger$, $W_y(0) = P\overline{W}_y(0)P^\dagger$, where $\overline{W}_x, \overline{W}_h, \overline{W}_y$ are diagonal matrices of eigenvalues. We substitute the diagonalized form, use $(PDP^\dagger)^\dagger = PD^*P^\dagger$ (where $D^*$ denotes the conjugate of $D$), and simplify to get:

$$\tau\frac{d}{dt_\theta}\overline{W}_x = \sum_{i=1}^{T} \overline{W}_h^{*(T-i)}\overline{W}_y^*(D_i - \overline{W}_y\overline{W}_h^{T-i}\overline{W}_x) \tag{179}$$

$$\tau\frac{d}{dt_\theta}\overline{W}_h = \sum_{i=1}^{T-1}\sum_{r=0}^{T-i-1} \overline{W}_h^{*(r)}\overline{W}_y^*(D_i - \overline{W}_y\overline{W}_h^{T-i}\overline{W}_x)\overline{W}_x^*\overline{W}_h^{*(T-i-1-r)} \tag{180}$$

$$\tau\frac{d}{dt_\theta}\overline{W}_y = \sum_{i=1}^{T}(D_i - \overline{W}_y\overline{W}_h^{T-i}\overline{W}_x)\overline{W}_x^*\overline{W}_h^{*(T-i)} \tag{181}$$

Let $a_\alpha, b_\alpha, c_\alpha, d_{\alpha,i}$ be the $\alpha^{\text{th}}$ diagonal entry of $\overline{W}_x, \overline{W}_h, \overline{W}_y, D_i$, respectively. We can rewrite the above equations in terms of these connectivity modes that decouple along eigenvalue dimensions $\alpha$.

$$\tau\frac{d}{dt_\theta}a_\alpha = \sum_{i=1}^{T} b_\alpha^{*(T-i)}c_\alpha^*(d_{\alpha,i} - c_\alpha b_\alpha^{T-i}a_\alpha) \tag{182}$$

$$\tau\frac{d}{dt_\theta}b_\alpha = \sum_{i=1}^{T-1}(T-i)c_\alpha^*(d_{\alpha,i} - c_\alpha b_\alpha^{T-i}a_\alpha)a_\alpha^*b_\alpha^{*(T-i-1)} \tag{183}$$

$$\tau\frac{d}{dt_\theta}c_\alpha = \sum_{i=1}^{T}(d_{\alpha,i} - c_\alpha b_\alpha^{T-i}a_\alpha)a_\alpha^*b_\alpha^{*(T-i)} \tag{184}$$

Finally, we can formulate an energy function by writing the gradient flow equations of the connectivity modes as the Wirtinger partial derivatives of the complex conjugates:

$$E = \frac{1}{2\tau}\sum_{i=1}^{T}(d_i - cb^{T-i}a)(d_i - cb^{T-i}a)^* \tag{185}$$

$$= \frac{1}{2\tau}\sum_{i=1}^{T}|d_i - cb^{T-i}a|^2 \tag{186}$$

$$\tau\frac{d}{dt_\theta}a = -\frac{\partial E}{\partial a^*} \tag{187}$$

$$\tau\frac{d}{dt_\theta}b = -\frac{\partial E}{\partial b^*} \tag{188}$$

$$\tau\frac{d}{dt_\theta}c = -\frac{\partial E}{\partial c^*} \tag{189}$$

$$\tag{190}$$

In the case where all terms are real, the energy function has an identical form to the SVD case considered in the main text. We make some initial investigations into the behavior of the connectivity modes and task dynamics in the complex plane in Appendix O, but note that this will be an important direction for future work.

## O. Learning dynamics of rotations in the complex plane

In this section, we study the learning dynamics of networks trained on rotational task dynamics (i.e., with complex eigenvalues) and show that our general framework extends naturally to this case. In particular, task dynamics are now represented by the data-correlation eigenvalues, which are potentially complex-valued but can still be decomposed into constant and dynamic components, such that $d_t = \delta f(\lambda, t)$. The task dynamics with 0-loss solutions (constant, inverse-exponential, and exponential) that we consider in the main text extend to this case and yield the same solutions as before, as shown in Figure A9.

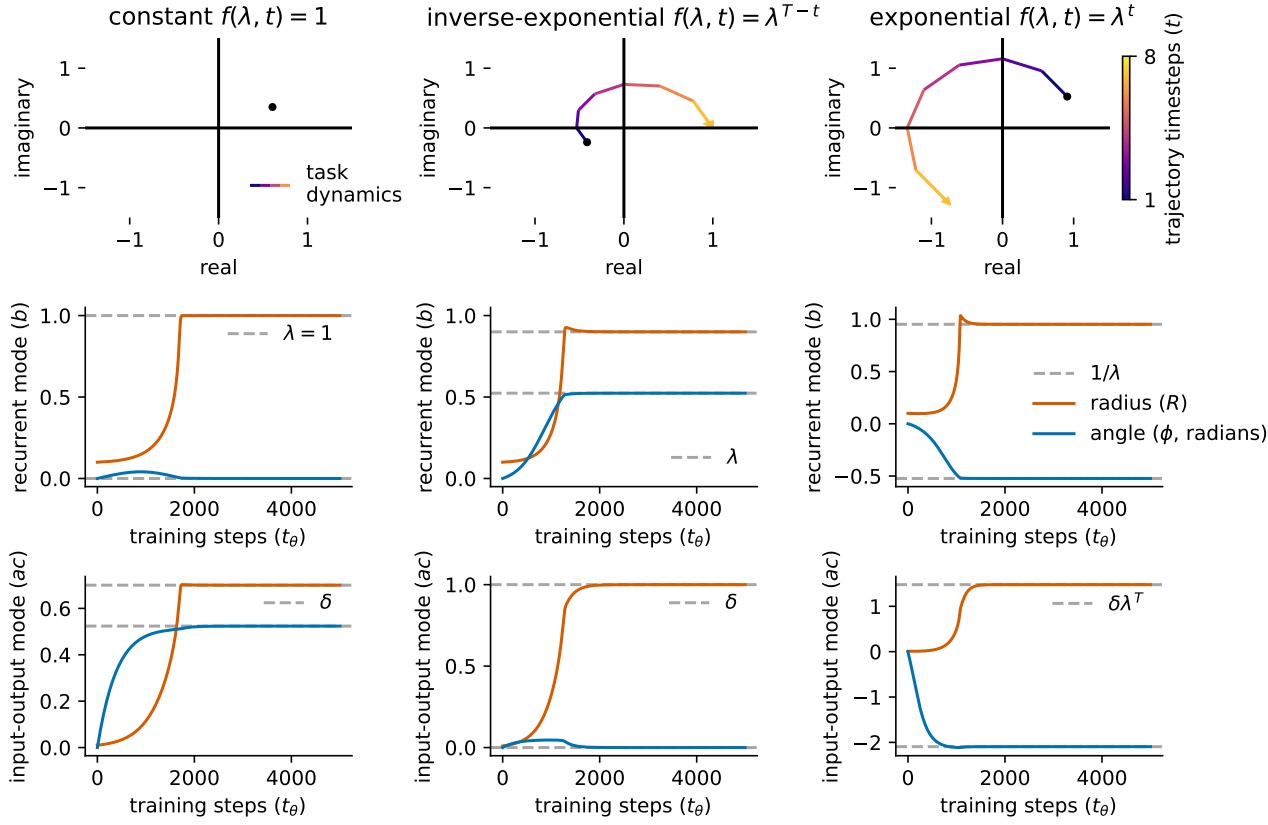

*Figure A9.* **Learning of different rotational task dynamics in the complex plane.** The task dynamics with zero-loss solutions that we consider in the main text extend naturally to rotational dynamics in the complex plane– (*left*) constant $f(\lambda, t) = 1$, (*middle*) inverse-exponential $f(\lambda, t) = \lambda^{T-t}$, and (*right*) exponential $f(\lambda, t) = \lambda^t$. (*Top row*) We plot examples of the task dynamics for the three cases in the complex plane. We keep the rotational angle the same across the different dynamics (constant $\phi_\delta = \pi/6$; inverse-exponential, exponential $\phi_\lambda = \pi/6$) and vary the radius to yield stable dynamics (constant $R_\delta = 0.7$, inverse-exponential $R_\lambda = 0.9$, exponential $R_\lambda = 1.05$). We plot the learning dynamics of (*middle row*) the recurrent mode $b$ and (*bottom row*) the input-output mode $ac$ in terms of their polar coordinates to show how the network learns (*orange*) the radius and (*blue*) angle components of the complex-valued data-correlation eigenvalues. In all cases, the different components of the connectivity modes converge to the predicted global optimums for the different task dynamics.

**Eigenvalues in polar coordinates.** More specifically, we can rewrite the complex eigenvalues in terms of polar coordinates in the complex plane such that $\delta = R_\delta e^{\phi_\delta i}, \lambda = R_\lambda e^{\phi_\lambda i}$. This means that inverse-exponential task dynamics correspond to $f(\lambda, t) = R_\lambda^{T-t} e^{(T-t)\phi_\lambda i}$, with the same global solutions as before at $b = \lambda = R_\lambda e^{\phi_\lambda i}, ac = \delta = R_\delta e^{\phi_\delta i}$. Similarly, exponential task dynamics are given by $f(\lambda, t) = R_\lambda^t e^{t\phi_\lambda i}$, with global solutions at $b = \frac{1}{\lambda} = \frac{1}{R_\lambda e^{\phi_\lambda i}}, ac = \delta\lambda^T = R_\delta e^{\phi_\delta i}(R_\lambda e^{\phi_\lambda i})^T$. Finally, constant task dynamics are given by $f(\lambda = 1, t) = 1$, with trivial solutions given by $b = \lambda = 1, ac = \delta = R_\delta e^{\phi_\delta i}$.

**Learning dynamics in polar coordinates.** One interesting observation is that the learning dynamics of different modes differ significantly when studied in the complex plane. While the learning of the radius is consistent with the dynamics in non-rotational LRNNs, the dynamics of the rotational angle differ substantially (Figure A9), including across different connectivity modes and task dynamics. In particular, the angle component begins adapting immediately at the onset of training, especially compared to the radial component. We suspect this behavior may be similar to the phenomena of eigenvector alignment which happens early in training (Atanasov et al., 2022), although in this case the eigenvalue angle is aligning rather than the eigenvectors themselves.

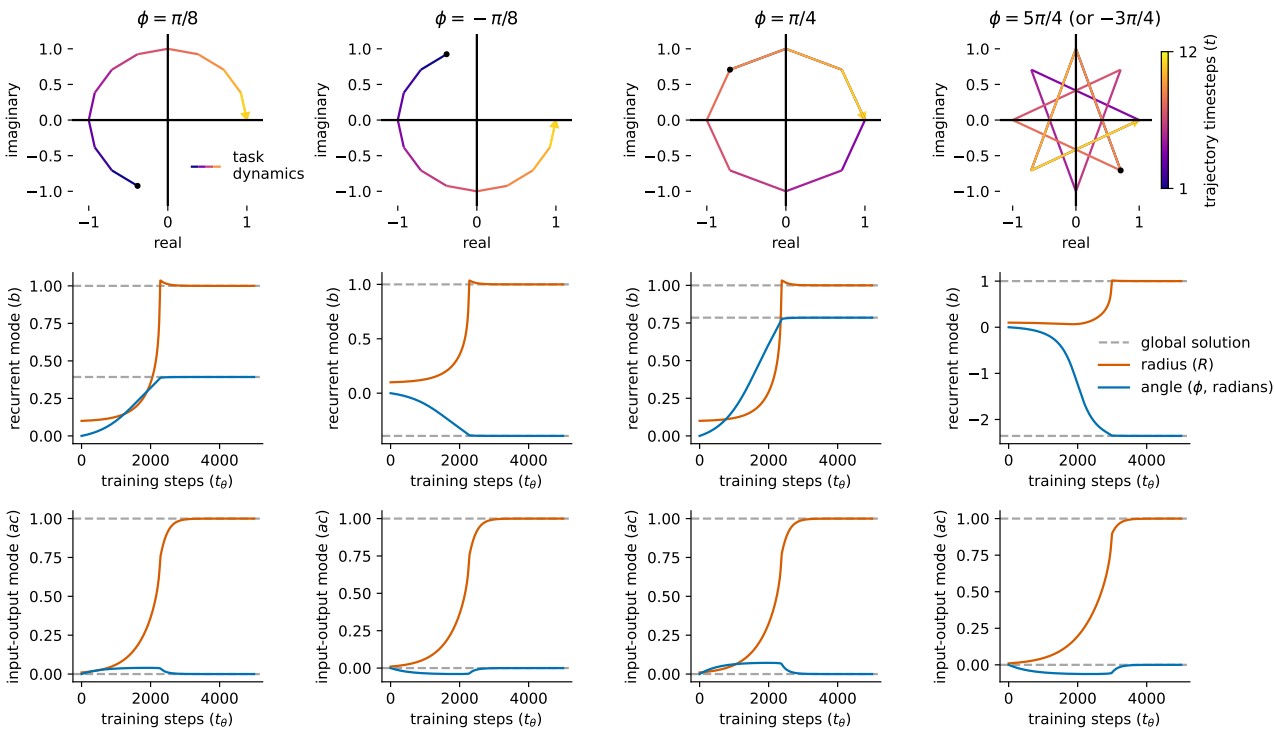

*Figure A10.* **Learning of inverse-exponential task dynamics with different rotational angles.** We study how different rotational angles $\phi$ impact task dynamics (*top row*) and learning dynamics ((*middle row*) recurrent mode, (*bottom row*) input-output mode), keeping the radius constant ($R = 1$) for $\phi = \pi/8, -\pi/8, \pi/4, -3\pi/4$. In the two leftmost columns, by studying complex conjugates, we can see that positive imaginary parts ($R\sin(\phi) > 0$) produce clockwise rotations, and negative imaginary parts produce counterclockwise rotations, and that these also correspond to the direction of perturbation in the input-output modes. In the two rightmost columns, we can see how increasing the rotational angle impacts the geometry of the task dynamics. We can also see that the mode trained on $\phi = 5\pi/4$ converges the slowest because its optimum lies the furthest from the initialization at $b(0) = 0.1$.

**Solution stability, extrapolation ability, and the ordering of learning.** One consequence of the direct extension of the task dynamics we consider and their solutions to the complex plane is that we can show that our results of task dynamics determining solution stability and extrapolation ability still hold. Networks trained on exponential task dynamics still fail to extrapolate because the global solution is dependent on trajectory length ($a^\star c^\star = \delta\lambda^T$). Furthermore, 'early-importance' task dynamics lead to unstable solutions, yielding eigenvalues greater than 1. In the case of complex eigenvalues, early-importance task dynamics correspond to dynamics where the radius decreases over the trajectory, also known as a *spiral sink*. Finally, the same principle of 'larger and later' eigenvalues being learned faster still applies in this setting. This is especially evident by comparing convergence speeds for different radii $R_\lambda, R_\delta$ (not shown), same as we do in Appendix E.

For simplicity, in the remainder of this section, we analyze the consequences of different $\lambda$ (setting $\delta = 1$) for inverse-exponential task dynamics ($d_t = \delta f(\lambda, t); f(\lambda, t) = \lambda^{T-t}$; i.e., a rotational LRNN teacher). Because only $\lambda$ is complex-valued in this case, we drop the subscripts on the radius and angle task dynamic parameters from hereon and simply refer to $\lambda$ as $\lambda = Re^{\phi i}$. Note however that different values of $\delta$ ($= R_\delta e^{\phi_\delta i}$) still impact task dynamics and learning dynamics, although we do not study this here.

**Impact of rotational angle.** We illustrate some examples of how changing different components of $\lambda$ — the angle $\phi$ (Figure A10) and the radius $R$ (Figure A11) — manifest in the task dynamics, the learning dynamics, and the network solutions. Recall that inverse-exponential task dynamics are given by $d_t = \delta\lambda^{T-t} = \delta R^{T-t}e^{(T-t)\phi i}$. The rotational angle is given by $\phi$. When the imaginary part of the solution ($\lambda$) is positive such that $R\sin(\phi) > 0$, the task dynamics rotate clockwise in the complex plane ($\phi = \pi/8$ in Figure A10). Conversely, when the imaginary part is negative ($R\sin(\phi) < 0$), the task dynamics rotate counterclockwise in the complex plane ($\phi = -\pi/8$ in Figure A10). In the case we study here, although the scaling component of the task dynamics $\delta$ is real-valued, the input-output modes still adapt along a (small)

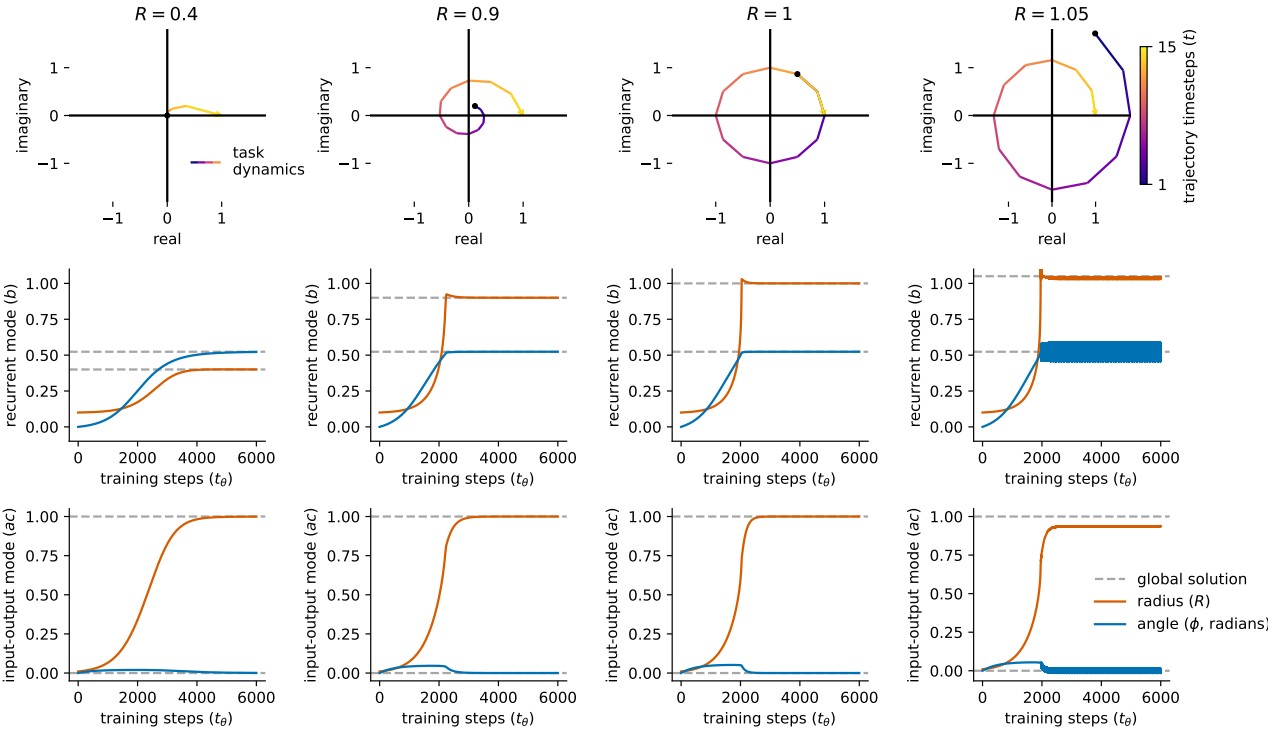

*Figure A11.* **Learning of inverse-exponential task dynamics with different radii.** We study how different radii $R$ impact task dynamics (*top row*) and learning dynamics ((*middle row*) recurrent mode, (*bottom row*) input-output mode), keeping the rotational angle constant ($\phi = \pi/6$) for $R = 0.4, 0.9, 1, 1.05$. First we can see that radii less than 1 ($R = 0.4, 0.9$) produce spiral source dynamics (or late-importance dynamics) where the radius increases through the trajectory of the task dynamics. Conversely, radii greater than 1 ($R = 1.05$) produce the opposite– spiral sinks or early-importance dynamics, which also lead to (*middle and bottom*) unstable solutions. Finally, radii equal to 1 ($R = 1$) produce a circular rotation/limit cycle in the task dynamics. Finally, by comparing small radii ($R = 0.4$) and large radii ($R = 1.05$), we can see that larger radii correspond to faster learning.

imaginary part before converging to a final real-valued solution. The sign of this imaginary part is determined by the sign of the imaginary part of the recurrent mode (i.e., both recurrent and input-output modes learn along either negative imaginary parts or both along positive imaginary parts). Furthermore, the magnitude of the imaginary deviation of the input-output modes is dependent on the magnitude of the angle ($\phi = \pi/4$ in Figure A10).

**Rotational angle affects learning speed.** By studying the complex plane, we can also see that learning speed is impacted by the distance from initialization to the global optimum and that the rotational angle influences this in a way that differs from only considering the magnitude of the eigenvalue/radius being learned. In particular, we know that trajectories with 'larger and later' eigenvalues/radii are learned faster (Appendix E). However, learning speed is also impacted by the distance from initialization to the solution, and in the complex plane this can be independent from the magnitude of the radius, which is especially clear when studying the effect of the rotational angle. For example, a set of modes all trained on inverse-exponential task dynamics with a radius of $R = 1$ will learn at different speeds depending on their rotational angle ($\phi$) relative to some fixed initialization. If the recurrent modes are all initialized at, say, $b(0) = 0.2$, modes with their solution further from initialization (as given by their rotational angle; for instance at $\phi = \pm 3\pi/4$) will learn slower, and modes with solutions closer to initialization (for instance at $\pm\phi = \pi/4$) will learn faster. On the other hand, if the network is initialized at $b(0) = -0.2$, this effect is reversed and the modes trained on task dynamics with a rotational angle $\phi = \pm 3\pi/4$ will learn faster and $\pm\phi = \pi/4$ will learn slower. This effect is subtly noticeable in Figure A10 comparing between $\phi = \pi/8$ versus $\phi = -3\pi/4$. However, we note that this effect mostly emerges if weights have large initializations. When initializing the network with small random weights, this effect is quite small and learning speed is mostly determined by the magnitude of the eigenvalues.

**Impact of radius.** The task dynamics we study in the main text correspond exactly to the case when the rotational angle is zero. In fact, some of the same phenomena apply here and are determined by the radius $R$. In the case of inverse-exponential dynamics, a radius less than 1 corresponds to what we call 'late-importance' task dynamics (where the data-correlation eigenvalues increase across the trajectory $|d_t| < |d_{t+1}|$), which manifests as task dynamics with increasing radius across the trajectory in the complex plane such that $R^{T-t} < R^{T-(t+1)}$, producing a *spiral source* ($R = 0.4, 0.9$ in Figure A11). When the radius is 1, we have a perfect rotation, or *limit cycle* ($R = 1$ in Figure A11). Finally, if the radius is greater than 1 and we have 'early-importance' task dynamics (data-correlation eigenvalues decrease across the trajectory $|d_t| > |d_{t+1}|$), the radius of the rotation $R^{T-t} > R^{T-(t+1)}$ (or the data-correlation eigenvalues) decreases across trajectory, producing a *spiral sink* ($R = 1.05$ in Figure A11).

As we noted earlier, we can see that early-importance task dynamics (or task dynamics that produce spiral sinks), have global solutions at $b > 1$, yielding unstable solutions ($R = 1.05$ in Figure A11). Similarly to before, we also see that learning speed is dependent on the radius size such that larger radii $R$ are learned faster than smaller ones ($R = 0.4$ versus $R = 1.05$ in Figure A11), again recapitulating a similar result to that of the largest singular values being learned first in Saxe et al. (2014).

# P. Simulations

Code for all simulations can be found at https://github.com/aproca/LRNN_dynamics

## P.1. LRNN initialization

### P.1.1. ALIGNED LRNN

To initialize aligned LRNNs, we reverse-engineer the weight matrices starting from the connectivity modes as described in Appendix B and Appendix N. We specify the initialization of the connectivity modes (input, recurrent, output) in each dimension as hyperparameters, which are the diagonal matrices $\overline{W}_x, \overline{W}_h, \overline{W}_y$.

**SVD** For the form based on SVD, we create orthogonal matrices $R_x$, $R_y = (R_x)^{-1}$ and use the left and right singular matrices $U_y, V_x$ of the data correlation matrices (see below in Appendix P.4.1) to form the weight matrices according to:

$$W_x = R_x \overline{W}_x V_x^\top \tag{191}$$

$$W_h = R_y \overline{W}_h R_x^\top \tag{192}$$

$$W_y = U_y \overline{W}_y R_y^\top \tag{193}$$

**Eigendecomposition** For the form based on an eigendecomposition, we use the eigenvectors of the data correlation matrices to form the real-valued weight matrices according to

$$W_x = P \overline{W}_x P^\dagger \tag{194}$$

$$W_h = P \overline{W}_h P^\dagger \tag{195}$$

$$W_y = P \overline{W}_y P^\dagger \tag{196}$$

### P.1.2. UNALIGNED LRNN

We create unaligned LRNNs by initializing the weights with a Gaussian distribution of mean 0 and standard deviation $\sigma/\sqrt{N_{\text{in}}}$, where $\sigma$ is a specified hyperparameter and $N_{\text{in}}$ is the row-size of the corresponding weight matrix.

## P.2. Training

Networks are trained using gradient descent on the mean squared error and automatic differentiation in order to validate our theoretical results. We modify our learning timescale to account for the additional scalars introduced by taking the mean over $P$ samples the and output dimension $N_y$ ($\tau = PN_y/\eta$, where $\eta$ is the learning rate) when comparing to simulation.

### P.3. Recovering connectivity modes

Recall that the learning dynamics are decoupled along singular value (or eigenvalue) dimensions ($\alpha$) for networks with aligned weights at initialization.

**SVD**  The matrices $R_y, R_x, U_y, V_x$ used for initializing LRNN weights will stay constant throughout training. To recover the connectivity modes, which drive learning in the network, at a particular time in training, we simply perform the inverse of the operation done at initialization,

$$\overline{W}_x(t_\theta) = R_x^\top W_x(t_\theta) V_x \tag{197}$$

$$\overline{W}_h(t_\theta) = R_y^\top W_h(t_\theta) R_x \tag{198}$$

$$\overline{W}_y(t_\theta) = U_y^\top W_y(t_\theta) R_y \tag{199}$$

When using networks that are not initialized with aligned weights, we just compute the singular values of each weight matrix.

**Eigendecomposition**  Similarly, the eigenvectors $P$ used for initializing LRNN weights will also stay constant throughout training. To recover the connectivity modes, we again perform the inverse of the operation done at initialization,

$$\overline{W}_x(t_\theta) = P^\dagger W_x(t_\theta) P \tag{200}$$

$$\overline{W}_h(t_\theta) = P^\dagger W_h(t_\theta) P \tag{201}$$

$$\overline{W}_y(t_\theta) = P^\dagger W_y(t_\theta) P \tag{202}$$

### P.4. Tasks

#### P.4.1. STRUCTURED TASK DYNAMICS

**SVD**  To create data with input-output correlation matrices that have constant left and right singular vectors and temporally-structured singular value dynamics, we similarly reverse-engineer the equations in Appendix B. We first generate random Gaussian input ($\boldsymbol{X}_{1:T}$) centered at 0, which is then whitened.

The data singular values $S_{1:T}$ are created by setting the singular values in each dimension $\alpha$ and at each trajectory timestep $t$ according to the specified task dynamics $f$ (constant: $f(\lambda_\alpha, t) = 1$, exponential $f(\lambda_\alpha, t) = \lambda_\alpha^t$, inverse-exponential $f(\lambda_\alpha, t) = \lambda_\alpha^{T-t}$) and hyperparameters $\delta_\alpha, \lambda_\alpha$, such that $s_{\alpha,t} = \delta_\alpha f(\lambda_\alpha, t)$.

We generate the left and right singular vectors $U_y, V_x$ by taking the SVD of a random matrix. Finally, we create the output according to

$$\boldsymbol{Y}_T = \sum_{i=1}^{T} U_y S_i V_x^\top \boldsymbol{X}_t \tag{203}$$

**Eigendecomposition**  We take a similar approach, except with an eigendecomposition. The eigenvalues $D_{1:T}$ are created by setting the (potentially complex-valued) eigenvalues in each dimension $\alpha$ and at each trajectory timestep $t$ according to the specified task dynamics $f(\lambda_\alpha, t)$.

To construct the data correlation matrices ($\Sigma^{YX_{1:T}}$) with constant eigenvectors ($P$) across time, we use a block-diagonal aproach (i.e., a real Schur form $\Sigma^{YX_t} = QHQ^{-1}$ where $H$ is upper or lower quasi-triangular and $Q$ is orthogonal). The eigenvalues are selected such that each complex eigenvalue has a complex conjugate in another dimension. For the eigenvalues at each trajectory timestep $D_t$, we construct a block diagonal matrix $H_t$, such that for each real eigenvalue there is a $1 \times 1$ block and for each pair of complex conjugate eigenvalues $\mu \pm i\nu$, there is a $2 \times 2$ block

$$\begin{bmatrix} \mu & -\nu \\ \nu & \mu \end{bmatrix}$$

Next, we construct the orthogonal matrix $Q$. For each real eigenvalue, we add as a column a one-hot vector with the value one in the dimension of the eigenvalue. For each complex eigenvalue block, we create complex column vectors: $[1, i]^\top$ for $\mu + i\nu$ and $[1, -i]^\top$ for $\mu - i\nu$ in their respective block dimensions and 0 elsewhere. Similar to the SVD case, we use the same constructed $Q$ across time, only changing the eigenvalues according to the task dynamics, such that $\Sigma^{YX_t} = QH_tQ^{-1}$. We create the output according to

$$Y_T = \sum_{i=1}^{T} QH_iQ^{-1}X_t \tag{204}$$

### P.4.2. MODIFIED TASK DYNAMICS

In Section 3.4, we study how changing the relationship between the correlations of input from the past $(1 : T - 1)$ and the input at the last timestep $(T)$ influences the solutions learned by connectivity modes. To do this, we simply create the singular values from $1 : T - 1$ as in Appendix P.4.1, but change the the singular values at the last timestep $T$ to some specified value.

For the Dirac delta task, we simply set the singular values at the first $(S_1)$ and last $(S_T)$ timesteps to some specified values, and set the rest of the singular values to 0.

### P.4.3. SENSORY INTEGRATION TASKS

To create the sensory-integration tasks, we use the multisensory integration task from Neurogym (Molano-Mazón et al., 2022) to generate stimuli in four input dimensions (removing the fixation input). We do not modify the data further (i.e., it is not whitened).

For the experiments on extrapolation ability, to create the target, we simply take the mean or the sum of the inputs along the trajectory in each dimension.

For the experiments on stability, we create the target by either computing $\boldsymbol{y}_T = 0.1 \sum_{t=1}^{T} t\boldsymbol{x}_t$ for late importance or $\boldsymbol{y}_T = \sum_{t=1}^{T} \frac{1}{t}\boldsymbol{x}_t$ for early importance.

