# OpenReview forum: "Learning dynamics in linear recurrent neural networks"
_ICML.cc/2025/Conference — ICML 2025 oral_

### Official Review · Reviewer_q5bu · 2025-03-12

**Overall Recommendation:** 4

**Summary:**

The authors study the learning dynamics of linear RNNs. This is similar to the  method in Saxe et al 2014, but extended to the recurrent setting. The extension leads to an energy function that has a sum of recurrent terms, and thus there is an interplay between recurrent and feedforward modes. The authors show the implication of the learning dynamics on what can and can’t be learned, and on the relative importance of late and early components of the task.

## After rebuttal

After reading all the rebuttals and discussion with all reviewers, I am keeping my score.

**Claims And Evidence:**

The claims are supported by both analytic derivations and numerical experiments.

**Essential References Not Discussed:**

The topic of lazy vs rich in recurrent networks is discussed in this paper:
Schuessler, Friedrich, Francesca Mastrogiuseppe, Srdjan Ostojic, and Omri Barak. “Aligned and Oblique Dynamics in Recurrent Neural Networks.” Edited by Tatyana O Sharpee and Michael J Frank. eLife 13 (November 27, 2024): RP93060. https://doi.org/10.7554/eLife.93060.

**Experimental Designs Or Analyses:**

No issues

**Methods And Evaluation Criteria:**

There are no benchmarks, but none are needed. The simulations performed are adequate for the problem at hand.

**Other Comments Or Suggestions:**

Line 298 “see” twice
Line 815 “more” appears twice.

**Other Strengths And Weaknesses:**

The extension of Saxe 2014 to RNNs provides new insights and explains phenomena that are unique to this setting.

**Questions For Authors:**

None

**Relation To Broader Scientific Literature:**

As explained in the paper, this work is related to learning dynamics in feedforward networks and to the evolution of kernels with learning.
The analysis of linear RNN with relation to data dynamics is novel.

**Theoretical Claims:**

I read all proofs, but did not verify every step of the math. The main derivation is similar to that of Saxe 2014. It would be useful to stress why the cross-term does not appear in this case. Is this because of the aligned assumption?

---

> ### Author Rebuttal · Authors · 2025-04-01
>
> We thank the reviewer for the positive feedback. In addition to answering the question about our main derivation and related work, we would like to point the reviewer to the list of new developments we have made since the initial submission, available in our rebuttal to Reviewer CWVm ([direct link here](https://openreview.net/forum?id=KGOcrIWYnx&noteId=RuV35U4JcB)). These include further mathematical and simulation results, which we hope will make our theory more rigorous and general.
>
> > The main derivation is similar to that of Saxe 2014. It would be useful to stress why the cross-term does not appear in this case. Is this because of the aligned assumption?
>
> Yes, the cross-term does not appear both because of the aligned assumption and because of the assumption that left and right singular vectors are constant across time, allowing us to diagonalize the weights such that each singular value dimension becomes decoupled. This can be thought of as rotating the weights so that they are aligned with the principal directions of the data at initialization. We note that other work (Braun et al., 2022) has alleviated the aligned assumption in feedforward networks but the derivation is quite involved and challenging to apply to recurrent networks as it is restricted to two-layer networks. In general, the aligned assumption appears to be reasonable when initializing a network with small random weights and has been shown to occur early in training in such cases (Atanasov et al., 2021). We will make sure to clarify this in-text.
>
> > The topic of lazy vs rich in recurrent networks is discussed in this paper: Schuessler, Friedrich, Francesca Mastrogiuseppe, Srdjan Ostojic, and Omri Barak. “Aligned and Oblique Dynamics in Recurrent Neural Networks.” Edited by Tatyana O Sharpee and Michael J Frank. eLife 13 (November 27, 2024): RP93060. https://doi.org/10.7554/eLife.93060.
>
> We thank the reviewer for pointing us to this interesting and important reference. We are now running additional rich and lazy learning experiments varying the scale of input-output connectivity modes in LRNNs with rotational dynamics to see whether similar phenomena might occur in the linear case.
>
> > Line 298 “see” twice Line 815 “more” appears twice.
>
> We thank the reviewer for pointing out the typo. We have fixed all typos in the new version of the manuscript.

---

### Official Review · Reviewer_zrFq · 2025-03-13

**Overall Recommendation:** 4

**Summary:**

The authors study the learning dynamics of linear recurrent neural networks (LRNNs). Using the approach of Saxe et al. (ICLR '14) to study deep linear feed-forward neural networks, the authors develop a similar theory for LRNNs. Under some assumptions on the weight matrices, the LRRN dynamics decouple into a set of uncorrelated modes that evolve independently.

The authors find that LRNNs learn with larger and later singular values first, akin to the faster learning of larger singular values by linear feed-forward networks (sec. 3.2). They consider a task generated by a matching linear RNN and are able to analyse the stability of training (i.e. the problem of vanishing / exploding gradients) and of extrapolation in this setting. Since their dynamical equations decompose into an equation for the recurrent mode, and for the input-output mode, they are also able to identify the emergence of low-rank structures in the weights (sec 3.4) They also find that the ''richness'' of the learning dynamics, as measured by the distance of the learnt network to its NTK at initialisation, increases as recurrent computations become more important. An numerical experiment with a sensory-integration experiment  where the assumptions are not exactly fulfilled shows good agreement with the theory regardless.


## After the discussion

I maintain my score.
I want to note that while I am aware with some of the recent literature on the theory of RNNs, I am much more familiar with the literature on the learning dynamics in feed-forward neural networks, so there might be some related literature I am missing.

**Claims And Evidence:**

The authors claim to introduce a new mathematical description of the learning dynamics of RNNs. While the assumption are somewhat restrictive (joint diagonalisability), they are stated clearly by the authors (kudos for that!) and they mirror the assumptions made by Saxe et al. and are therefore likely reasonable to make progress on this problem. In the feedforward case, the theory based on these assumptions has had quite an impact on subsequent research, so I think this is a good contribution (and long overdue!) to be done for linear RNNs, too. The wealth of phenomena that they analyse further suggests that their solution of linear RNNs is a useful toy model to understanding recurrent neural networks.

**Essential References Not Discussed:**

None that I know of

**Experimental Designs Or Analyses:**

Not applicable.

**Methods And Evaluation Criteria:**

This is a theory paper, and its methods are appropriate.

**Other Comments Or Suggestions:**

See above

**Other Strengths And Weaknesses:**

This is a nice paper that provides a clean theoretical analysis of linear recurrent neural networks, which remain an important tool to study sequential data, and remain the most important neural network model for theoretical neuroscience. Given the huge impact of similar work on linear feed-forward neural networks, I think this framework will prove a valuable tool for theory, too, and I think it should be accepted at ICML.

**Questions For Authors:**

No further questions.

**Relation To Broader Scientific Literature:**

The authors do seem to acknowledge related literature appropriately, both in the introduction and the discussion at the end. however, while I am aware with some of the recent literature on the theory of RNNs, I am much more familiar with the literature on the learning dynamics in feed-forward neural networks, so there might be some related literature I am missing.

**Theoretical Claims:**

I looked at the derivation of the dynamical equations, and it seemed all good to me. Assumptions for this analysis are clearly spelled out at the top of page 3.

---

> ### Author Rebuttal · Authors · 2025-04-01
>
> We thank the reviewer for the positive feedback and the encouraging words. In addition to one clarification about this review below, we would like to point the reviewer to the list of new developments we have made since the initial submission, available in our rebuttal to Reviewer CWVm ([direct link here](https://openreview.net/forum?id=KGOcrIWYnx&noteId=RuV35U4JcB)). These include further mathematical and simulation results, which we hope will make our theory more rigorous and general.
>
> > The authors claim to introduce a new mathematical description of the learning dynamics of RNNs. While the assumption are somewhat restrictive (joint diagonalisability), they are stated clearly by the authors (kudos for that!) and they mirror the assumptions made by Saxe et al. and are therefore likely reasonable to make progress on this problem. In the feedforward case, the theory based on these assumptions has had quite an impact on subsequent research, so I think this is a good contribution (and long overdue!) to be done for linear RNNs, too. The wealth of phenomena that they analyse further suggests that their solution of linear RNNs is a useful toy model to understanding recurrent neural networks.
>
> We complete agree with the reviewer that this is the beginning of a longer research direction. While the assumption of joint diagonalizability has been commonly used in the learning dynamics literature (Saxe et al., 2014; 2018) and there is evidence that the parameters of the network align in this way at the beginning of training (Atanasov et al., 2021) (see also our response to Reviewer q5bu), we agree that this assumption may be restrictive in certain cases. Luckily, even if our assumptions are somewhat restrictive, our empirical results seem to be a bit more general, such as in the case of our sensory integration tasks.

---

### Official Review · Reviewer_9zFA · 2025-03-13

**Overall Recommendation:** 4

**Summary:**

While there has been substantial progress in understanding the learning dynamics of feedforward networks, there is relatively less work on studying it in the context of recurrent networks, especially when considering the task dynamics as well. In this paper, the authors analyze the learning dynamics of a linear RNN and derive an analytical solution with certain assumptions on the data and model initialization, for which the motivate well. Using this solution, they show that task dynamics impact the stability of the solution and the network's ability to extrapolate in time. Specifically, they divide the task into time-independent and time-varying components and show how these different aspects influence the solution. They also show that there is a trade-off between computation being more recurrent or feedforward, and that there is a phase transition between these two modes in terms of the task dynamics. This specifically emerges in the case where the data is not perfectly learnable, such as when the last time step, for which the loss is computed, does not follow the task dynamics of the rest of the data trajectory. Moreover, the trade-off between these types of computations leads to low-rank solutions that are known to emerge during RNN training. This is due to an effective regularization term that pops out of the energy function. The authors also derive a neural tangent kernel (NTK) for finite-width linear RNNs. Using this NTK, they show that recurrence leads to rich learning. Finally, they use their theory to explain how linear RNNs learn to perform sensory integration tasks, which is a common paradigm in computational neuroscience.

**Claims And Evidence:**

Overall, the paper did a good job supporting its claims with a thorough and extensive evaluation. One point of confusion for me is whether some of these results hinge on the fact that we only compute the loss on the final time step. This seems especially relevant for the phase transition between feedforward and recurrent computation. This does not make or break the study, but is an important point to address in the text when trying to understand these results.

**Essential References Not Discussed:**

A relevant paper that talks about rich and lazy learning in nonlinear RNNs that may be worth citing includes:
- Payeur et al., "Neural Manifolds and Learning Regimes in Neural-Interface Tasks," bioRxiv 2023.

**Experimental Designs Or Analyses:**

I looked over the experimental design of the main results and found them to be sound.

**Methods And Evaluation Criteria:**

The analyses performed on the linear RNN and assumptions on task dynamics make sense given the scope of the study. It is also interesting to see how well the theory matches simulations in the sensory integration task, which is of broader relevance to the computational neuroscience community. It was especially interesting to see how the choice of objective (mean- vs. sum-integration) impacted extrapolation performance.

**Other Comments Or Suggestions:**

I think the results in Section 3.4 could be better explained, as it took me awhile to understand what was going on. Figure 4, in particular, is hard to parse, and I need more hand-holding when going through it.

**Other Strengths And Weaknesses:**

The paper is well written and does a good job supporting  its claims. It is a novel approach to studying learning dynamics in linear RNNs, which is relevant to both deep learning theory and computational neuroscience.

**Questions For Authors:**

Does this trade-off between feed-forward and recurrent computation partially depend on the fact that the last time step is all that's included in your objective function? How would these results change if intermediate time steps are also penalized?

**Relation To Broader Scientific Literature:**

The authors do a great job situating their work in the broader related literature. Most relevant to their work is the body of literature on analyzing deep linear networks and neural tangent kernels, for which they cite important papers. Prior work which has looked at linear RNNs has not accounted for the impact of the task dynamics on learning dynamics and the solutions. The results in this paper also have bearing on the emergence of low-rank connectivity in RNNs, which is a common framework in computational neuroscience to account for the observation that neural activity tends to be low-dimensional in many brain regions. This work also bears on extrapolation to longer sequences, which is of primary concern in RNNs in general.

**Theoretical Claims:**

I skimmed the proofs in Appendix A of the gradient flow equations and energy functions and Appendix J which extends their approach to multiple outputs over time. Nothing popped out to me as incorrect.

---

> ### Author Rebuttal · Authors · 2025-04-01
>
> We thank the reviewer for the positive feedback and the clear summary of our work. In addition to addressing the limitations identified in this review, we would like to point the reviewer to the list of new developments we have made since the initial submission, available in our rebuttal to Reviewer CWVm ([direct link here](https://openreview.net/forum?id=KGOcrIWYnx&noteId=RuV35U4JcB)). These include further mathematical and simulation results, which we hope will make our theory more rigorous and general.
>
> > Overall, the paper did a good job supporting its claims with a thorough and extensive evaluation. One point of confusion for me is whether some of these results hinge on the fact that we only compute the loss on the final time step. This seems especially relevant for the phase transition between feedforward and recurrent computation. This does not make or break the study, but is an important point to address in the text when trying to understand these results.
>
> We thank the reviewer for the positive feedback and for raising this important and nuanced issue. In the main text and results, we consider the case where only the output at the last timestep enters into the loss for simplicity and interpretability. We fully agree with the reviewer that it is important to consider more general cases and how our results generalize. In the supplementary material, we have made an effort to generalize our derivation of the energy function and gradient flow equations to the case where the loss is computed over the output at each timestep. Crucially, we were able to derive the functional form of the energy function for this (multi-output) case, and it only differs from the original (single-output) case by an additional summation over the singular values of the data correlations for different output timesteps.
>
> An intuitive interpretation of this is that the multi-output energy function contains a summation of the single-output energy function for different outputs.  Although this can result in different solutions depending on the task dynamics as a result of optimizing for many different outputs, we would expect several of our main results to hold if task dynamics are consistent across different outputs (e.g., in the case of a teacher-student setup). We are currently working to generalize some of our other results to this setting, including our derivation of the NTK and variation of feedforward and recurrent computation (see response below). However, it is possible that there may be additional phenomena to account for when considering the relationship between output dynamics, which will be an important direction for future work.
>
> > A relevant paper that talks about rich and lazy learning in nonlinear RNNs that may be worth citing includes:
> > Payeur et al., "Neural Manifolds and Learning Regimes in Neural-Interface Tasks," bioRxiv 2023.
>
> We thank the reviewer for pointing us to this relevant reference and we will make sure to include in our discussion of related work.
>
> > I think the results in Section 3.4 could be better explained, as it took me awhile to understand what was going on. Figure 4, in particular, is hard to parse, and I need more hand-holding when going through it.
>
> We thank the reviewer for the valuable feedback and we apologize for the lack of clarity on our part. We will revise Section 3.4 and modify Figure 4 to make the results easier to follow.
>
> > Does this trade-off between feed-forward and recurrent computation partially depend on the fact that the last time step is all that's included in your objective function? How would these results change if intermediate time steps are also penalized?
>
> We thank the reviewer for the insightful question. Following your question, we’ve now simulated this in the Dirac delta task and observed that pruning of connectivity modes still exists based on the magnitudes of $s_1$ and $s_T$, similar to the single-output case, although we have not yet proved mathematically that it’s a proper phase transition. We will make sure to include this and more analyses for the case where the loss includes outputs over multiple timesteps in the manuscript.

---

### Official Review · Reviewer_CWVm · 2025-03-13

**Overall Recommendation:** 3

**Summary:**

This paper analyzes the learning dynamics encountered when training a linear recurrent neural network (i.e., a linear time invariant system)  using gradient descent. The paper derives a reduced form of the learning dynamics, and connects the stability of the model to the task being trained on.

**Claims And Evidence:**

The paper is reasonably clear mathematically, although I found it rather difficult to follow what was being said much of the time in the main text.

**Essential References Not Discussed:**

This paper seems quite close in approach and spirit: https://arxiv.org/pdf/2407.07279

Also, there does not appear to be any discussion of the recent popular work on State Space Models (SSMs), which are LTI systems: https://arxiv.org/abs/2008.07669

**Experimental Designs Or Analyses:**

Yes, the experiments corresponding to Figs 4 and 5. No issues encountered.

**Methods And Evaluation Criteria:**

Yes.

**Other Comments Or Suggestions:**

N/A

**Other Strengths And Weaknesses:**

I think the paper would benefit from much clearer writing. There are too many distracting inline equations and symbols and the results are written in what is (in my opinion) an overly technical way.

**Questions For Authors:**

What separates your approach from that of Saxe (2014, 2018)? Is it just the depth of the network you study?

**Relation To Broader Scientific Literature:**

The paper is of interest to neuroscience researchers, who have long been using recurrent neural networks as models of the brain. The paper is also of interest to people in deep learning working on SSMs, which are essentially linear (either time-invariant or time-varying) dynamical systems.

**Theoretical Claims:**

I went through the derivation of equations 5-8 and I did not find any issues.

---

> ### Author Rebuttal · Authors · 2025-04-01
>
> We appreciate the positive feedback on the relevance and potential of the work for the neuroscience and machine learning community. These comments will enable substantial improvement to the manuscript.
>
> In addition to addressing the limitations below, we would like to highlight additional results and developments we have made to the paper since the time of initial submission, which we hope will substantially improve our paper by making our theory more rigorous and general. In particular, our new results cover:
> * **Learnability**: We prove that task dynamics that are exponential-in-time (i.e., constant, exponential, and inverse-exponential) are the only task dynamics with 0-loss solutions in our model.
> * **Phase transition**: Using Landau theory, we show analytically that for Dirac delta task dynamics, there is a (first-order for $T>3$, second-order for $T=3$) phase transition, solely dependent on the ratio of the first and last singular value. We verify these results in simulation.
> * **Experimental results**: We extend our sensory-integration tasks to also illustrate how early-importance task dynamics lead to instability.
> * **Generalisation**: We perform additional experiments to show that our feature-learning result generalizes to RNNs learning tasks with rotational dynamics (non-constant singular vectors through time).
> * **Learning dynamics**: We extend our solution to the recurrent connectivity modes to an analytical approximation using the Faa di Bruno formula.
>
> We also thank the reviewer for identifying the limitations of our paper, which we address below:
>
> > I think the paper would benefit from much clearer writing. There are too many distracting inline equations and symbols and the results are written in what is (in my opinion) an overly technical way
>
> We thank the reviewer for the valuable feedback and apologize for the lack of clarity. This work was written as a theory paper, and as such we have tried to condense the most critical equations into the main text and left the remainder of the derivations to the appendix. However, we agree that theory is more useful when it is clear and can be widely understood. To enhance clarity, we will revise the writing to be clearer by describing concepts in simple terms and reducing the overuse of symbols in-text where possible.
>
> > The paper is of interest to neuroscience researchers, who have long been using recurrent neural networks as models of the brain. The paper is also of interest to people in deep learning working on SSMs, which are essentially (either time-invariant or time-varying) dynamical systems. This paper seems quite close in approach and spirit: https://arxiv.org/pdf/2407.07279
> >
> > Also, there does not appear to be any discussion of the recent popular work on State Space Models (SSMs), which are LTI systems: https://arxiv.org/abs/2008.07669
>
> We thank the reviewer for pointing us to these important references, which are certainly related to our work (especially Smekal 2024). We did not discuss SSMs aside from citing a few references, but they are certainly very relevant to the study of RNNs and an interesting and important extension for future work. We will make sure to include the citations and more discussion around this topic, including the new results provided by our framework over and above those in these references (such as the results on stability and extrapolation, phase transition, rich and lazy learning, and validation in a sensory task).
>
> > What separates your approach from that of Saxe (2014, 2018)? Is it just the depth of the network you study?
>
> Because the RNN receives sequential input that is related to the output in a time-dependent way, the energy function includes a summation over time which includes time-dependent singular values. In contrast, in Saxe et al. (2014, 2018), there is no time-dependence in the data (or sequence) and thus only a single set of singular values.
>
> More explicitly, two aspects where the difference between our work and Saxe et al. (2014, 2018) can be seen clearly are in the data specification and in the energy function. Regarding the data, Saxe et al. (2014, 2018) considers the case where the data correlation has N singular values (one for each dimension), whereas we consider T data correlations, yielding T x N singular values in total. This helps us draw conclusions that are unique to the RNN setup (like the fact that later singular values are learned first), which would have been impossible to obtain by just extending the depth of Saxe et al.’s model. Regarding the energy function, Saxe et al. (2014, 2018) have an energy function given by $E=(s-ca)^2$, whereas our energy function is given by $E=\sum_{i=1}^T (s_i – cb^{T-i}a)^2$. The sum and the power of $b^{T-i}$ also enable some RNN-specific conclusions (such as the phase transition governing recurrent and feedforward computations), which again would not be possible to study with deep feedforward networks.

---

### Decision · Program_Chairs · 2025-05-01

**Decision:**

Accept (oral)

**Comment:**

The paper investigates the learning dynamics of linear recurrent neural network (LRNN) and present clear math results. All reviewers expressed excitement of the theoretical results and appreciate the identified learning dynamics of LRNNs. So I recommend acceptance of this paper.